# Regulating Anatomy-Aware Rewards via Trajectory-Integral Feedback for Volumetric Computed Tomography Analysis

Tianwei Lin [* 1 2]   Zhongwei Qiu [* 2 3 1]   Jie Cao [1]   Jiang Liu [1]   Wenjie Yan [1]   Bo Zhang [4]   Yu Zhong [1]
Wenqiao Zhang [† 1]   Yingda Xia [2]   Ling Zhang [† 2]

## Abstract

Medical vision-language models (VLMs) have rapidly advanced as general-purpose multimodal assistants, yet their deployment in 3D Computed Tomography (CT) analysis remains constrained by a persistent mismatch between optimization objectives and clinical rigor. Current Reinforcement Learning (RL) paradigms still rely on lexical proxy signals that induce "*Evaluation Hallucinations*", where models optimize linguistic fluency rather than factual clinical correctness, leading to diagnostically critical errors. To bridge this gap, we introduce the **Clinical Abnormality Benchmarking Substrate (CABS)**, a structured system that decomposes radiology reports into verifiable clinical semantic units. Using CABS, we identify a "*Mechanistic Divergence*" in standard RL, where surface-similarity rewards drive policy gradients to bypass medical facts. We therefore propose **Trajectory-Integral Feedback GRPO (TIF-GRPO)**, a novel framework integrating control-theoretic principles into policy optimization. By formulating clinical reasoning as a pseudo-temporal trajectory for anomaly discovery, TIF-GRPO regulates anatomy-aware rewards via an integral feedback loop that penalizes persistent omissions as cumulative state errors and suppresses hallucinations as excessive control effort. Experiments on 3D CT benchmarks demonstrate that our approach significantly enhances abnormality detection and clinical faithfulness, establishing a new paradigm for fine-grained regulation in medical VLMs. Our project is available at GitHub.

[*]Equal contribution [†]Corresponding authors. [1]Zhejiang University [2]DAMO Academy, Alibaba Group [3]Hupan Lab [4]University of Electronic Science and Technology of China. Correspondence to: Wenqiao Zhang <wenqiaozhang@zju.edu.cn>, Ling Zhang <ling.z@alibaba-inc.com>.

*Proceedings of the 43rd International Conference on Machine Learning*, Seoul, South Korea. PMLR 306, 2026. Copyright 2026 by the author(s).

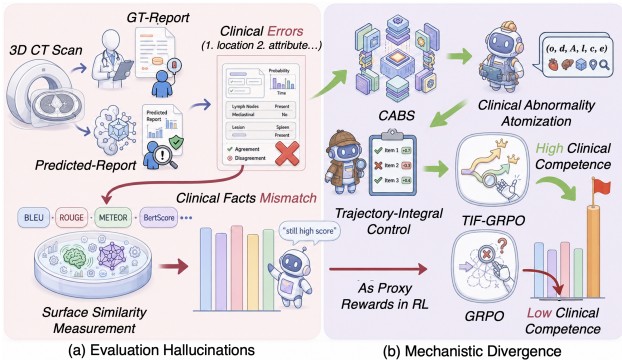

*Figure 1.* Overview of the "*Evaluation Hallucinations*" and "*Mechanistic Divergence*". **(a)** Surface-similarity proxy signals induce evaluation hallucinations, where high-scoring predictions mismatch GT clinical facts. **(b)** Our CABS framework enables accurate abnormality-level measurement, and TIF-GRPO applies trajectory-integral control based on CABS to suppress hallucinations and align optimization with clinical fidelity.

## 1. Introduction

While the 2D slice-level Computed Tomography (CT) offers a foundational view, 3D volumetric CT analysis provides superior clinical utility by transcending the inherent limitations of a fragmented planar view (Qiu et al., 2022; Xin et al., 2025; Lee et al., 2024; Lin et al., 2025). By preserving voxel-level spatial topology, 3D CT analysis enables high-fidelity reconstruction of lesion-tissue associations—a prerequisite for precise surgical and diagnostic interventions. To harness this complex spatial data, recent 3D medical vision-language models (VLMs) have emerged, showing transformative potential as AI radiological imaging expert for 3D CT analysis (Hamamci et al., 2024c; Jiang et al., 2025; Bai et al., 2024; Lin et al., 2026). However, existing research largely inherits the general-purpose training paradigm for medical domain adaptation. This direct transfer often overlooks a critical clinical boundary: unlike general domains, **Factuality** and **Faithfulness** are the non-negotiable baselines of medical analysis. In clinical reporting, diagnostically decisive findings are often sparsely embedded within otherwise routine and semantically homogeneous descriptions; as a result, objectives driven primarily

by distributional fitting may overemphasize frequent stylistic regularities while underweighting the small set of clinically critical signals, so that even subtle hallucinations can introduce disproportionate diagnostic risk and undermine diagnostic integrity.

This oversight leads to a dual crisis in the current medical VLM paradigm: **(i) evaluation hallucinations**, where metrics are misaligned with clinical capability, and **(ii)** the **misleading nature of RL rewards** built upon these hallucinatory proxy signals. Existing evaluations rely heavily on lexical overlap (e.g., BLEU (Papineni et al., 2002), ROUGE (Lin, 2004), METEOR (Banerjee & Lavie, 2005)) or simplistic scoring by coarse-grained medical semantic similarity (e.g., RadGraph (Jain et al., 2021), RaTEScore (Zhao et al., 2024), BioBert (Lee et al., 2020)). These proxy signals establish an evaluation essentially detached from high-stakes medical diagnosis, manifesting as a systematic divergence between evaluation scores and clinical competence. As shown in Fig 1 (a), samples scoring high on these metrics often commit fatal clinical errors, such as misinterpreting pathological attributes or mismatching anatomical laterality. More critically, as shown in Fig 1 (b), when RL algorithms (Pan et al., 2025; Lai et al., 2025; Fan et al., 2025; Dai et al., 2025) utilize rewards based on superficial accuracy, models fall into Reward Hacking, mimicking the style of ground-truth answers at the expense of true image understanding. This indicates that without a clinically rigorous reward, policy optimization may actively drive models away from medical facts.

To correct this goal misalignment, we advocate a paradigm shift in evaluation: from treating medical VLM outputs as unstructured text to interpreting them as compositions of discrete, verifiable clinical facts. Consequently, we propose the **Clinical Abnormality Benchmarking Substrate (CABS)**, a structured representation framework that decomposes radiology reports into atomic abnormality units grounded in clinical ontology. CABS explicitly encodes each abnormality as a tuple of organ, pathological entity, anatomical location, attributes, diagnostic certainty, and supporting textual evidence, thereby establishing a precise and auditable foundation for clinical factuality. This decomposition enables comprehensive evaluation of a model's ability to detect and describe abnormalities through organ- and finding-level precision, recall, and F1 scores, providing a faithful measure of true clinical competency. While prior efforts such as RadGraph (Jain et al., 2021) and RaTEScore (Zhao et al., 2024) also structure free-text reports, they ultimately rely on semantic similarity between extracted fragments, which remains a proxy disconnected from ground-truth pathology. Multiple radiologists validated CABS with 98.6% approval across key dimensions, confirming its reliability as a unified substrate for both evaluation and training.

Utilizing CABS, we systematically deconstruct the alignment behaviors induced by heterogeneous reward signals in existing medical RL methods. We identify a phenomenon of **Mechanistic Divergence**: surface-similarity rewards often fail to reliably discriminate between clinically accurate and inaccurate outputs within a response group, leading optimization dynamics to diverge from true clinical fidelity. Furthermore, when driven by sparse clinical feedback, standard RL algorithms face severe instability, often collapsing into safe modes that ignore long-tail abnormalities to maximize average lexical scores. To address these challenges, we propose **Trajectory-Integral Feedback GRPO (TIF-GRPO)**, a novel RL framework that integrates control-theoretic principles into Group Relative Policy Optimization (GRPO) (Shao et al., 2024). Unlike standard RL methods (Pan et al., 2025; Dai et al., 2025), which react only to instantaneous rewards, TIF-GRPO reformulates the clinical reasoning process as a **pseudo-temporal trajectory**. It regulates anatomy-aware rewards through an integral feedback loop that adaptively penalizes persistent diagnostic omissions (False Negatives) as cumulative state errors and suppresses speculative hallucinations (False Positives) as excessive control effort. Our approach grounds the model's reasoning in the rigorous logic of clinical practice, stabilizing policy gradients and prioritizing diagnostic factuality.

Our contributions can be summarized as follows:

- **Metric:** We propose **CABS**, a system that decomposes medical VLM outputs into clinical abnormality units, unifying the evaluation standards for clinical detection and description.

- **Analysis:** We provide a systematic analysis of existing medical similarity-based RL methods and characterize the **Mechanistic Divergence** phenomenon, demonstrating how proxy rewards drive policies away from clinical fact correctness.

- **Methodology:** We introduce **TIF-GRPO**, a novel framework that leverages trajectory-integral control to stabilize policy gradients and provide fine-grained regulation of clinical anatomy-aware reward signals.

- **Results:** Our TIF-GRPO achieves state-of-the-art (SOTA) performance across multiple 3D CT benchmarks, significantly improving clinical factuality and faithfulness.

## 2. Related Work

Recent medical vision-language models (VLMs) have evolved from cross-modal alignment systems to increasingly unified and reasoning-capable architectures. Early efforts such as BioMedGPT (Zhang et al., 2023), LLaVA-Med (Li

et al., 2023a), and Med-Flamingo (Moor et al., 2023) established multimodal adaptation pipelines for medical understanding, while more recent models such as HealthGPT (Lin et al., 2025), MedGemma (Sellergren et al., 2025), Hulu-Med (Jiang et al., 2025), and TumorChain (Li et al., 2026) further expanded diagnostic reasoning and generalist perception. In the 3D domain, RadFM (Wu et al., 2025a), CT2Rep (Hamamci et al., 2024b), Med3DInsight (Chen et al., 2024c), M3D-LaMed (Bai et al., 2024) and Om-niCT (Lin et al., 2026) progressively advanced volumetric CT interpretation and report generation. In parallel, evaluation of medical report generation has moved beyond pure lexical overlap metrics such as BLEU and ROUGE toward more structured or semantically informed frameworks, including RadGraph (Jain et al., 2021), RaTEScore (Zhao et al., 2024), ReEvalMed (Li et al., 2025a), and related LLM-assisted protocols; however, these metrics still function largely as proxy signals and often remain weakly grounded in image-anchored clinical facts. Reinforcement learning has likewise become an important route for clinical refinement, from general algorithms such as PPO (Schulman et al., 2017), RLOO (Ahmadian et al., 2024), and GRPO (Shao et al., 2024) to domain-specific medical variants such as MedVLM-R1 (Pan et al., 2025), Med-R1 (Lai et al., 2025), ChestX-Reasoner (Fan et al., 2025), and QoQ-Med (Dai et al., 2025). Nevertheless, current medical RL pipelines typically inherit reward designs derived from surface similarity, sparse rubric scores, or weak clinical feedback, leaving a gap between optimization dynamics and true diagnostic fidelity. Our work is positioned at the intersection of these three threads: we replace proxy-style evaluation with a clinically grounded substrate and use it to define a trajectory-level RL objective aligned with factual clinical reasoning. A more detailed review of these three directions is provided in Appendix A, including medical VLMs (Appendix A.1), medical report evaluation (Appendix A.2), and reinforcement learning in medical VLMs (Appendix A.3).

## 3. Methodology

### 3.1. Preliminary and Problem Formulation

Let $\mathcal{V}$ denote the 3D CT volumes, where a specific input is represented as $V \in \mathbb{R}^{Z \times H \times W}$, where Z, H, W represent the size of 3D CT. Given a clinical task prompt $q$ (e.g., *"Analyze the findings in this CT scan"*), the Large VLM, parameterized by $\theta$, aims to generate a clinically faithful response $y = \{w_1, \ldots, w_n\}$. Formally, the model learns a conditional policy $\pi_\theta(y|V, q) = \prod_{t=1}^{n} P(w_t|V, q, w_{<t})$. Existing medical VLMs usually follow a two-stage paradigm for domain adaptation: STF and RL.

**Supervised Fine-Tuning (SFT):** Maximizing the expected log-likelihood of ground-truth reports $y_{gt}$ under the model $\pi_\theta$ via minimizing the loss $\mathcal{L}_{SFT} =$

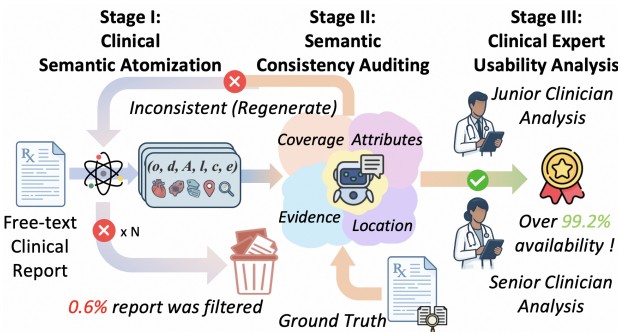

*Figure 2.* Overview of the CABS workflow. Free-text clinical reports are converted into structured clinical semantics, followed by semantic consistency auditing and clinician usability analysis, achieving an overall acceptance ratio of approximately $99.4\% \times 99.2\% \approx 98.6\%$.

$-\mathbb{E}_{(V,q,y_{gt}) \sim \mathcal{D}}[\sum_{t=1}^{T} \log \pi_\theta(y_{gt}|V, q, w_{<t})]$, where $\mathcal{D}$ denotes the training dataset.

**Reinforcement Learning (RL):** To further align the model with clinical factuality, Group Relative Policy Optimization (GRPO) is used as a typical RL strategy to optimize the model. For each prompt $(V, q)$, GRPO samples a group of $G$ outputs $\{y_1, y_2, \ldots, y_G\}$ from the old policy $\pi_{\theta_{old}}$. The set of rewards is defined as $R_G = \{r1, r2, ..., r_G\}$. GRPO computes the normalized group advantage: $A_i^{GRPO} = \frac{r_i - \mu_G}{\sigma_G + \epsilon}$, where $\mu_G$ and $\sigma_G$ are the mean and standard deviation of $R_G$. $\epsilon$ denotes a small constant to avoid division by zero. With clip strategy and KL divergence penalty, the optimization objective is:

$$\mathcal{J}(\theta) = \mathbb{E}_{(V,q) \sim \mathcal{D}}[\frac{1}{G} \sum_{i=1}^{G} \hat{A}_i - \beta D_{KL}(\pi_\theta || \pi_{ref})],$$

$$\hat{A}_i(\theta) = min(\lambda_i \cdot A_i^{GRPO}, \text{clip}(\lambda_i, 1 - \epsilon, 1 + \epsilon) \cdot A_i^{GRPO}), \quad (1)$$

where $\lambda_i = \frac{\pi_\theta(y|V,q)}{\pi_\theta^{old}(y|V,q)}$ represents the importance ratio between policy model $\pi_\theta$ and old policy model $\pi_\theta^{old}$. $D_{KL}$ is the Kullback–Leibler divergence between the policy model and the reference model. $G$ is group number, $\epsilon$ is a small constant, $\beta$ is the weight of KL loss.

RL serves as a pivotal methodology for bolstering the performance of medical VLMs in downstream scenarios plagued by data scarcity, such as the identification of rare pathologies. However, traditional RL objectives typically optimize for superficial lexical similarity or rely on sparse clinical feedback. This reliance creates a misalignment known as *"Evaluation Hallucination"* (Discussed in Section 4.4), where high reward scores are achieved through linguistic mimicry rather than genuine clinical factuality. More critically, this reliance induces a phenomenon we term *"Mechanistic Divergence"* (Discussed in Section 4.5): under surface-similarity rewards, models fail to reliably discriminate between clinically accurate and inaccurate outputs within a response group, causing

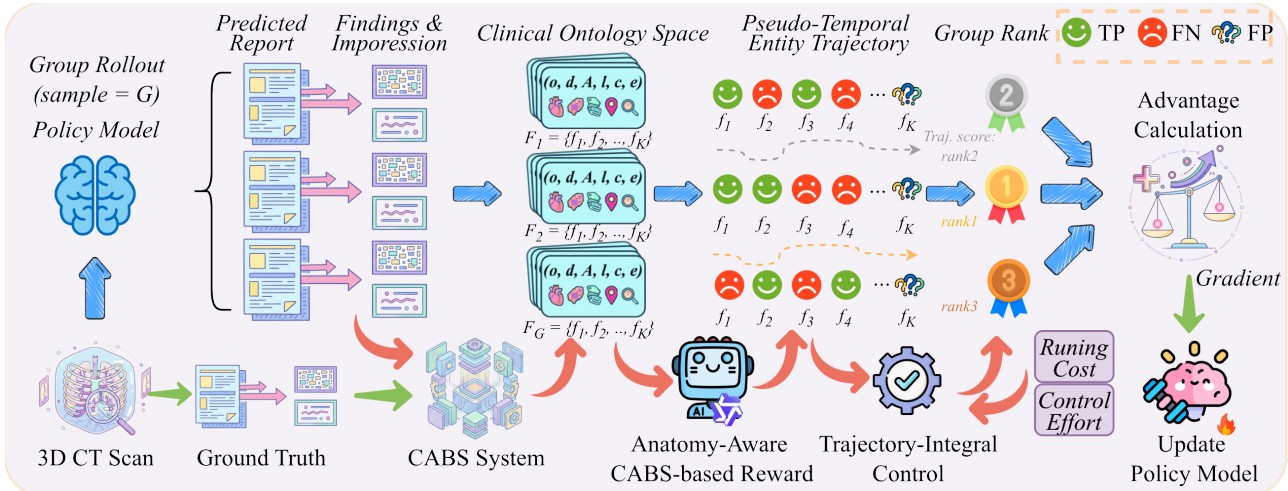

*Figure 3.* TIF-GRPO leverages CABS to decompose reports into clinical abnormality units, enabling trajectory-integral control that penalizes false positives and omissions for factuality-aligned RL.

the policy optimization process to diverge from true clinical fidelity.

To resolve this misalignment and ground policy optimization in clinical factuality, we propose the Clinical Abnormality Benchmarking Substrate, a structured evaluation system that decomposes reports into verifiable clinical units, enabling direct measurement of diagnostic fidelity.

### 3.2. Clinical Abnormality Benchmarking Substrate

To bridge the gap between linguistic fluency and clinical factuality, we propose the **Clinical Abnormality Benchmarking Substrate (CABS)**. Unlike traditional metrics that treat model outputs as monolithic strings, CABS provides a principled framework to decompose a free-text response $y$ into a set of $K$ discrete, verifiable clinical semantic units $\mathcal{F} = \{f_1, f_2, \ldots, f_K\}$. The workflow of CABS is shown in Fig 2. Formally, we define a clinical abnormality $f_i$ as a structured tuple within a predefined clinical ontology space $\mathcal{S}$:

$$f_i = \langle o, d, \mathcal{A}, l, c, e \rangle \in \mathcal{S}, \tag{2}$$

where:

- $o \in \mathcal{O}$ represents the target organ (e.g., *liver, lung*).
- $d$ denotes the abnormal entity (e.g., *cystic lesion*).
- $\mathcal{A} = \{a_1, a_2, \ldots, a_m\}$ is a set of pathological attributes describing morphology, density, and size (e.g., *14mm × 20mm, low-density*).
- $l$ denotes the anatomical location precision (e.g., *upper right lobe, liver S8*).
- $c$ indicates the diagnostic certainty (e.g., *definite, suspicious*).
- $e$ is the textual evidence extracted from the original report that grounds the abnormality.

We define a mapping function $\Phi : \mathcal{Y} \to \mathcal{P}(\mathcal{S})$ that trans-

forms the unstructured response space $\mathcal{Y}$ into the power set of clinical facts. For a generated response $y_{gen}$ and a reference report $y_{ref}$, the alignment is no longer measured by token-level overlap, but by the Semantic Fact Intersection over $\mathcal{S}$. Specifically, for a given organ $o$, we can compute the anatomy-level and abnormality-level precision, recall, and F1 scores, rather than lexical metrics like ROUGE. This structured substrate serves as the foundation for our subsequent trajectory-integral reward modulation.

### 3.3. TIF-GRPO

To achieve fine-grained control over diverse anatomical structures and mitigate the "*Mechanistic Divergence*" observed in traditional RL, we propose **Trajectory-Integral Feedback GRPO (TIF-GRPO)**. This framework is shown in Fig 3, which shifts the optimization objective from surface-level linguistic alignment to the regulation of a clinical reasoning trajectory under TIF control.

**Anatomy-Aware CABS-based Reward:** Unlike monolithic reward functions, we define a structured reward that decomposes the feedback into clinical semantic units $\mathcal{F}$. Specifically, for each clinical semantic unit $f_i \in \mathcal{F}$, we compute an individual hit reward $r_i$ based on the alignment between the generated output $y_{gen}$ and the specific anatomical/pathological constraints of $f_i$:

$$r_i = \text{hit}_i \cdot \left( l_i \cdot d_i + \frac{1}{2} \cdot |l_i - d_i| \right), \tag{3}$$

where $\text{hit}_i \in \{0, 1\}$ is a binary indicator denoting whether the model's prediction hits (correctly identifies) a relevant concept in the target report. $l_i \in \{0, 1\}$ and $d_i \in \{0, 1\}$ denote whether the model correctly identifies the location and abnormal entity, respectively. This CABS-based reward mechanism can better capture genuine clinical signals,

which is crucial for guiding the model toward a correct anatomical foundation.

The global instantaneous reward for the response is then formulated as the average intensity across all $K$ units in the reference trajectory $\mathcal{F} = (f_1, \ldots, f_K)$: $R_{CABS} = \frac{1}{K}\sum_{i=1}^{K} r_i$. By explicitly mapping predictions to these discrete anatomical states $r_i$, the framework transforms the high-entropy generation task into a continuous monitoring of clinical coverage. This decomposition provides a high-fidelity, anatomy-sensitive signal that prevents the policy from being misled by linguistic fluency at the expense of diagnostic factuality.

**Trajectory-Integral Control:** Building upon the CABS substrate, we introduce a novel Trajectory-Integral Reward that dynamically tracks the model's progression through the clinical reasoning space. Instead of evaluating the final output alone, we integrate the reward signal over the sequence of predicted abnormalities, penalizing both false negatives (missed findings) and false positives (hallucinated findings) in a graded, cumulative manner.

$$R_{TIF} = \alpha - \underbrace{\frac{\alpha}{K}\sum_{k=1}^{K}\left(1 - \frac{1}{k}\sum_{i=1}^{k} r_i\right)^2}_{\text{Running Cost (FN Integral)}}$$

$$+ \underbrace{\gamma\left(1 - \left(\frac{FP}{M+\varepsilon}\right)^2\right)}_{\text{Control Effort (FP Penalty)}} \quad (4)$$

$$+ \underbrace{\frac{1}{K}\sum_{i=1}^{K} r_i}_{\text{Terminal Reward}} + \underbrace{0.05 \cdot \mathbf{1}[M > 0]}_{\text{Exploration Bonus}},$$

where $K$ is the number of clinical semantic units in $y_{ref}$, and the **Terminal Reward** term denotes the current CABS reward. $\alpha$ and $\gamma$ are scalar hyperparameters that balance the trade-off between cumulative false-negative penalties and false-positive suppression. $\mathbf{1}[\cdot]$ is indicator function. $M$ represents the number of the abnormal entity predicted by the model, and the **Exploration Bonus** term encourages the model to explore anomalies. $FP$ represents the number of false positives, and the **Control Effort** term penalizes discouraging excessive false positives, analogous to the energy cost in classical control systems. This term rewards models that make precise, evidence-based assertions rather than speculative or redundant findings. The **Running Cost** term integrates the false-negative error over the sequence of clinical findings, with a squared penalty that emphasizes early and persistent misses. This design mimics the integral component in PID control (Johnson & Moradi, 2005) along the generative trajectory of the model's anomaly detection, ensuring the model prioritizes timely and reliable detection of critical abnormalities rather than achieving high recall

through late or inconsistent predictions. In **Running Cost** term, $1 - \frac{1}{k}\sum_{i=1}^{k} r_i$ simulates the FN errors accumulation at $k^{th}$ clinical semantic unit during the generation trajectory.

The TIF mechanism reformulates the RL objective as a fine-grained control problem. By mapping discrete clinical findings onto a sequential trajectory, TIF effectively steers the policy gradient to prioritize diagnostic factuality over superficial linguistic fluency. This cybernetic approach grounds the model's reasoning in the rigorous logic of clinical practice by adaptively penalizing persistent diagnostic omissions and suppressing speculative hallucinations. Importantly, this mechanism is fundamentally different from conventional reward shaping. Standard reward shaping usually applies pointwise numerical transformations to local rewards, without changing how credit is assigned across the response trajectory. In contrast, TIF-GRPO introduces a trajectory-integral reward that turns optimization from a simple summation of abnormality-level contributions into a path-dependent objective. This structural change modifies group-wise advantage ranking within GRPO, thereby altering the resulting gradient directions during policy updates. By explicitly injecting trajectory-level clinical signals into credit assignment, TIF-GRPO steers RL optimization toward clinically meaningful trade-offs rather than merely rescaling reward magnitudes.

**Boundary Cases**: **(i)** No ground-truth abnormalities ($K = 0$): The reward is dominated by the control effort term, penalizing any false positives and encouraging the model to remain silent about abnormalities. **(ii)** Model makes no predictions ($M = 0$): FP penalty is minimal, but if $K > 0$, both running cost and terminal reward are zero, discouraging complete silence when abnormalities exist.

**Optimization Objective:** TIF-GRPO integrates the trajectory-regulated rewards into the GRPO framework. For each sampled response $y_i$ in the group $G$, we calculate the unified TIF reward $R_{TIF,i}$ as defined in the Equation 4. The Trajectory-Regulated Advantage $\hat{A}_i^{TIF}$ is then derived by performing group-relative normalization:

$$A_i^{TIF} = \frac{R_{TIF,i} - \mu_G(R_{TIF})}{\sigma_G(R_{TIF}) + \epsilon}, i \in [1, G]. \quad (5)$$

By substituting the traditional advantage in Equation 1 with $A_i^{TIF}$, the final optimization objective of TIF-GRPO is to maximize:

$$\mathcal{J}^{TIF}(\theta) = \mathbb{E}_{(V,q)\sim\mathcal{D}}\left[\frac{1}{G}\sum_{i=1}^{G} \hat{A}_i^{TIF} - \beta D_{KL}(\pi_\theta||\pi_{ref})\right],$$

$$\hat{A}_i^{TIF}(\theta) = min(\lambda \cdot A_i^{TIF}, \text{clip}(\lambda, 1 - \epsilon, 1 + \epsilon) \cdot A_i^{TIF}), \quad (6)$$

where $\beta = 0.04$.

# 4. Experiments

## 4.1. Experimental Setting

**Baselines and Benchmarks.** We validate our method on Qwen3-VL-4B (Yang et al., 2025), and the training procedure follows a standard two-stage paradigm, consisting of Supervised Fine-Tuning and Reinforcement Learning. We compare general VLMs (Qwen3-VL (Yang et al., 2025), InternVL3.5 (Wang et al., 2025), Gemini-3-pro (Google DeepMind, 2025) and GPT-5 (OpenAI, 2025)) and medical-specific VLMs (RadFM (Wu et al., 2025a), M3D-LaMed (Bai et al., 2024), CT-CHAT (Hamamci et al., 2024a), Hulu-Med (Jiang et al., 2025), Fleming-VL (Shu et al., 2025)) with TIF-GRPO on four benchmarks: CT-RATE-Report, AMOS-MM-Report, CT-RATE-MCQ, and AMOS-MM-MCQ constructed from CT-Rate (Hamamci et al., 2024c) and AMOS-MM (Ji et al., 2022) dataset. We further evaluate our method on the MIMIC-CXR-Report task (Johnson et al., 2019) to investigate its cross-modal generalization capability.

**Metrics.** CT-RATE-Report and AMOS-MM-Report are free-text report generation tasks, while CT-RATE-MCQ and AMOS-MM-MCQ are multiple-choice question tasks. For evaluation, we use both traditional surface-similarity metrics (BLEU, ROUGE, METEOR, RadGraph, RaTEScore, BioBert) and comprehensive clinical metrics of the CABS system in three aspects: Entity Core (Precision, Recall, and F1 Score), Clinical Fidelity (Location Accuracy, Attribute Accuracy, and Fully-Consistent Accuracy), and Organ Coverage (Or-Rate and FMOr-Rate). The metrics in the CABS system measure the real clinical competence verified by multiple radiologists, and the specific meaning and calculation process of them are shown in Appendix C.

## 4.2. SOTA Results on Multiple Benchmarks

Tables 1 and 2 present a systematic evaluation of TIF-GRPO on CT-RATE-Report and AMOS-MM-Report. The results show that TIF-GRPO achieves SOTA performance across most key metrics on both benchmarks, and consistently outperforms general-purpose VLMs as well as existing medical-specific models on clinically relevant dimensions, including entity-level accuracy, clinical fidelity, and organ coverage. These findings indicate that TIF-GRPO leads to substantive improvements in aligning visual evidence with clinical semantics. Further analysis suggests that the observed performance gains are not attributable to stronger fitting of supervised data. Compared with CT-CHAT, our two-stage training pipeline uses only approximately 45% of its report training data, while CT-CHAT additionally leverages hundreds of thousands of SFT samples to supervise attribute and location signals. Under this setting, the model trained with SFT alone underperforms CT-CHAT; however, after introducing the TIF-GRPO RL stage, the model exhibits consistent and significant improvements across all metrics, ultimately surpassing CT-CHAT. This controlled comparison demonstrates that TIF-GRPO effectively guides the optimization of clinically relevant objectives, enabling substantial performance gains under limited supervision. In addition, we observe that CABS-based clinical metrics provide stronger discriminative power for evaluating CT report generation capabilities across models, whereas surface-level similarity metrics show limited separation among strong models, with relatively weak correlation between the two. This observation suggests that certain proxy-based automatic evaluation signals may insufficiently capture clinically meaningful differences in model behavior. We further analyze the implications of these signals and their influence on model optimization in subsequent sections.

To mitigate potential biases induced by model behavior preferences in free-form generation, we construct clinically grounded MCQ evaluation tasks under the CABS framework using CT-RATE-Report and AMOS-MM-Report (see Appendix B.1 for details), which explicitly constrain the model's action space and enable a more direct assessment of clinical discriminative capability. As shown in Table 3, TIF-GRPO achieves the best performance across all CABS-based clinical metrics, consistently outperforming both general-purpose and medical-specific models on the Existence, Location, and Attribute subtasks as well as in overall average score, while also exhibiting strong cross-dataset generalization, i.e., models trained on CT-RATE-MCQ (or AMOS-MM-MCQ) remain optimal when evaluated on the other dataset. Moreover, compared with GRPO variants that rely solely on accuracy-based rewards, TIF-GRPO further improves downstream MCQ performance by explicitly preserving TIF control (fine-grained clinical consistency constraints), suggesting that maintaining fine-grained clinical semantic signals during RL is critical for learning stable and generalizable medical representations beyond surface-level similarity.

We further emphasize that the generalizability of TIF-GRPO does not depend on a CT-specific ontology itself, but on two broader conditions: **(i)** the task can be decomposed into verifiable intermediate semantic units, and **(ii)** these units can be incorporated into trajectory-level credit assignment. Under these conditions, TIF-GRPO provides a unified optimization framework for tasks that previously lacked clinically meaningful RL objectives. To validate this claim, we extend our method to a 2D chest X-ray report domain and observe the same trend: as shown in Table 4, TIF-GRPO still yields substantial gains over SFT-only, ROUGE-based GRPO, and LLM-based GRPO with rubric reward, improving F1 from 25.3 to 70.8. This cross-modal result suggests that the benefit of TIF-GRPO arises from its trajectory-level clinical credit assignment mechanism based on CABS, rather than from assumptions specific to 3D CT anatomy alone.

*Table 1.* Comparison of general and medical-specific VLMs across entity, clinical, and organ metrics on CT-RATE-Report. Note that CT-CHAT[†] is trained on the full CT-RATE dataset.

| Model | #Params | Entity Core | | | | Clinical Fidelity | | | Organ Coverage | | Surface-Similarity | | |
|---|---|---|---|---|---|---|---|---|---|---|---|---|---|
| | | Precision | Recall | F1 | Acc | Loc-Acc | Att-Acc | FC-Rate | Or-Rate | FMOr-Rate | ROUGE | RaTEScore | Biobert |
| **General Large Vision Language Models** | | | | | | | | | | | | | |
| **Qwen3-VL-8B** | 8B | 13.73 | 9.80 | 11.42 | 84.14 | 56.18 | 29.75 | 3.08 | 11.66 | 1.75 | 16.06 | 52.74 | 73.90 |
| **InternVL3.5-8B** | 8B | 23.62 | 4.61 | 7.70 | 37.92 | 55.63 | 30.41 | 0.74 | 3.33 | 0.18 | 16.95 | 54.60 | 74.83 |
| **Gemini-3-Pro** | * | 24.58 | 18.96 | 21.37 | 81.46 | 63.71 | 41.06 | 6.26 | 17.85 | 3.93 | 17.10 | 54.91 | 75.02 |
| **GPT-5** | * | 39.71 | 19.94 | 26.26 | 73.79 | 69.87 | 47.28 | 8.33 | 18.48 | 5.09 | 13.08 | 50.73 | 74.15 |
| **Medical-Specific Large Vision Language Models** | | | | | | | | | | | | | |
| **RadFM** | 14B | 13.29 | 4.15 | 6.30 | 36.00 | 41.18 | 30.36 | 0.52 | 3.77 | 0.30 | 7.10 | 35.50 | 68.74 |
| **M3D-LaMed** | 7B | 8.77 | 5.98 | 7.02 | 85.17 | 35.56 | 18.76 | 0.56 | 6.47 | 0.23 | 8.39 | 38.74 | 69.58 |
| **CT-CHAT**[†] | 8B | 23.91 | 29.24 | 26.03 | 87.53 | **78.04** | 51.67 | 14.98 | 32.43 | 10.34 | 31.23 | 64.53 | 80.52 |
| **Hulu-Med** | 7B | 20.45 | 10.16 | 12.53 | 47.31 | 61.45 | 33.58 | 3.19 | 9.81 | 1.98 | 13.96 | 53.44 | 73.37 |
| **Fleming-VL-8B** | 8B | 20.52 | 2.34 | 4.18 | 29.22 | 49.29 | 24.63 | 0.47 | 1.56 | 0.27 | 16.26 | 55.33 | 73.91 |
| **Ours w/ SFT** | 4B | 34.59 | 12.85 | 18.73 | 61.45 | 67.99 | 52.22 | 5.68 | 12.06 | 2.71 | 37.09 | 66.88 | 82.26 |
| **Ours w/ TIF-GRPO** | 4B | 40.93 | 37.80 | 39.11 | 87.28 | 70.23 | **65.79** | 20.37 | **40.84** | 12.69 | **42.18** | **70.86** | 83.46 |
| **Ours w/ TIF-GRPO** | 8B | **41.24** | **38.21** | **39.67** | **87.66** | 71.06 | 64.40 | **20.67** | 40.51 | **13.57** | 41.04 | 70.26 | **83.67** |

*Table 2.* Comparison of general and medical-specific VLMs across entity, clinical, and organ metrics on AMOS-MM-Report.

| Model | #Params | Entity Core | | | | Clinical Fidelity | | | Organ Coverage | | Surface-Similarity | | |
|---|---|---|---|---|---|---|---|---|---|---|---|---|---|
| | | Precision | Recall | F1 | Acc | Loc-Acc | Att-Acc | FC-Rate | Or-Rate | FMOr-Rate | ROUGE | RaTEScore | Biobert |
| **General Large Vision Language Models** | | | | | | | | | | | | | |
| **Qwen3-VL-8B** | 8B | 6.08 | 4.95 | 5.44 | 59.88 | 59.26 | 25.68 | 0.89 | 5.33 | 0.55 | 17.92 | 48.60 | 75.29 |
| **InternVL3.5-8B** | 8B | 11.66 | 1.54 | 2.71 | 20.64 | 52.34 | 23.59 | 0.21 | 1.43 | 0.04 | 19.03 | 48.82 | 76.04 |
| **Gemini-3-Pro** | * | 10.26 | 10.48 | 10.33 | 87.24 | 55.60 | 40.97 | 3.09 | 11.46 | 1.99 | 19.07 | 47.51 | 75.77 |
| **GPT-5** | * | 15.15 | 10.05 | 12.08 | 76.13 | 75.13 | 44.17 | 3.65 | 9.99 | 2.58 | 17.18 | 47.17 | 75.15 |
| **Medical-Specific Large Vision Language Models** | | | | | | | | | | | | | |
| **RadFM** | 14B | 4.64 | 2.41 | 3.17 | 48.49 | 38.57 | 20.52 | 0.30 | 2.91 | 0.19 | 11.65 | 33.54 | 68.63 |
| **M3D-LaMed** | 7B | 6.14 | 5.75 | 5.93 | 84.77 | 35.79 | 24.13 | 0.69 | 6.94 | 0.39 | 11.22 | 35.25 | 65.10 |
| **CT-CHAT** | 8B | 1.33 | 0.60 | 0.82 | 70.51 | 69.44 | 11.11 | 0.02 | 0.31 | 0.00 | 18.03 | 40.58 | 72.47 |
| **Hulu-Med** | 7B | 10.40 | 8.09 | 7.60 | 44.31 | 55.93 | 37.26 | 1.74 | 7.97 | 1.16 | 16.56 | 49.96 | 74.85 |
| **Fleming-VL-8B** | 8B | 17.49 | 0.27 | 0.53 | 7.34 | 55.00 | 10.00 | 0.00 | 0.23 | 0.00 | 17.90 | 44.91 | 75.02 |
| **Ours w/ SFT** | 4B | 19.79 | 15.93 | 17.65 | 87.59 | 59.80 | 47.76 | 7.23 | 18.93 | 5.38 | 31.75 | 59.79 | **82.73** |
| **Ours w/ TIF-GRPO** | 4B | 24.17 | 41.22 | 30.47 | 95.06 | 76.80 | 57.77 | 22.77 | **49.95** | **17.91** | 29.04 | **58.53** | 81.54 |
| **Ours w/ TIF-GRPO** | 8B | **25.32** | **43.93** | **32.12** | **95.34** | **77.09** | **60.03** | **24.52** | 49.57 | 16.19 | 27.35 | 57.27 | 80.26 |

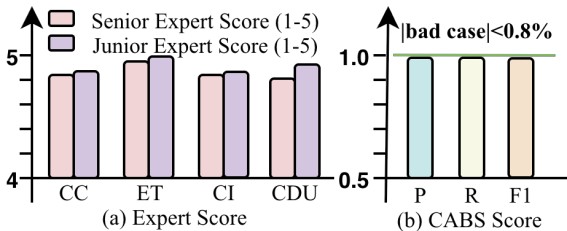

*Figure 4.* Clinical Competence Analysis of CABS System.

### 4.3. Clinical Competence Analysis of CABS System

CABS serves as a key anchor for validating both the existence of evaluation hallucinations and the effectiveness of TIF-GRPO. To this end, we conduct a systematic validation of CABS from the perspective of clinical capability analysis, combining assessments from clinical experts and large-model self-evaluations. Specifically, we evaluate the usability of CABS for abnormal unit decomposition across four evaluation dimensions (CC: Clinical Correctness, ET: Evidence Traceability, CI: Coverage Integrity, and CDU: Clinical Decomposition Usability in Appendix D).

As shown in Fig 4 (a), although senior clinical experts apply more stringent criteria, CABS still achieves an average score above 4.8 on the most rigorous CDU dimension, indicating that it provides high-quality and practically usable abnormal unit decompositions in the majority of cases. Furthermore, as shown in Fig 4 (b), we perform a consistency check by treating the original reports themselves as evaluation targets, in order to assess the consistency of CABS in both decomposition and evaluation under the abnormal unit specification. The results show that fewer than 0.8% of samples exhibit discrepancies of two or more abnormal units, further demonstrating the robustness and stability of CABS with respect to decomposition granularity and evaluation standards. In a nutshell, CABS-based indicators have been endorsed by clinicians and demonstrate superior alignment with actual clinical competence.

### 4.4. Evaluation Hallucination Analysis

We conduct a Spearman's rank correlation analysis on the results of over 10 models (e.g., Qwen3-VL, GPT-5) from Table 1, using two evaluation suites: **(i)** 6 conventional surface-similarity metrics, and **(ii)** 9 CABS-based clinical

*Table 3.* Performance comparison and Out-Of-Distribution evaluation on CT-RATE-MCQ and AMOS-MM-MCQ benchmarks.

| Model | #Params | CT-RATE-MCQ | | | | AMOS-MM-MCQ | | | | Total Avg. |
|---|---|---|---|---|---|---|---|---|---|---|
| | | Exist. | Loc. | Attr. | Avg. | Exist. | Loc. | Attr. | Avg. | |
| **General Large Vision Language Models** | | | | | | | | | | |
| Qwen3-VL-8B | 8B | 52.89 | 46.84 | 61.35 | 53.24 | 56.10 | 19.74 | 61.65 | 48.02 | 50.63 |
| InternVL3.5-8B | 8B | 51.18 | 45.15 | 63.77 | 52.47 | 54.10 | 17.98 | 58.74 | 45.88 | 49.18 |
| Gemini-3-Pro | * | 46.89 | 56.54 | 80.19 | 56.97 | 47.32 | 53.16 | 80.68 | 56.42 | 56.70 |
| GPT-5 | * | 53.53 | 56.96 | 83.57 | 61.25 | 50.89 | 51.59 | 80.61 | 59.82 | 60.54 |
| **Medical-Specific Large Vision Language Models** | | | | | | | | | | |
| RadFM | 14B | 51.82 | 28.69 | 26.09 | 39.96 | 52.55 | 22.81 | 27.67 | 39.10 | 39.53 |
| M3D-LaMed | 7B | 47.54 | 53.16 | 63.29 | 52.58 | 35.46 | 43.86 | 66.50 | 44.86 | 48.72 |
| CT-CHAT | 8B | 49.25 | 55.27 | 66.67 | 54.77 | 51.45 | 45.18 | 67.96 | 53.67 | 54.22 |
| Hulu-Med | 7B | 52.18 | 58.76 | 81.09 | 60.46 | 50.33 | 37.47 | 78.58 | 53.59 | 57.03 |
| Fleming-VL-8B | 8B | 52.89 | 55.27 | 65.70 | 56.42 | 52.11 | 38.60 | 64.08 | 51.41 | 53.92 |
| GRPO w/ Acc Reward (CT-RATE) | 4B | 55.03 | 73.00 | 88.41 | 67.29 | 53.46 | 53.51 | 85.44 | 59.89 | 63.59 |
| GRPO w/ Acc Reward (AMOS-MM) | 4B | 54.18 | 61.18 | 73.91 | 60.48 | 57.83 | 68.86 | 91.26 | 67.34 | 63.91 |
| TIF-GRPO w/ Acc Reward (CT-RATE) | 4B | **59.96** | **73.84** | **89.37** | **70.25** | 58.06 | 56.14 | 86.41 | 63.05 | 66.65 |
| TIF-GRPO w/ Acc Reward (AMOS-MM) | 4B | 55.03 | 68.35 | 85.99 | 65.53 | **66.13** | **76.32** | **95.15** | **74.24** | **69.67** |

*Table 4.* Performance of TIF-GRPO on MIMIC-CXR-Report.

| Training Strategy | Precision | Recall | F1 |
|---|---|---|---|
| **Baseline** | 21.1 | 33.1 | 25.3 |
| **SFT** | 44.1 | 21.2 | 28.6 |
| **GRPO+ROUGE** | 52.6 | 39.3 | 45.3 |
| **GRPO+LLM** | 60.9 | 46.7 | 52.7 |
| **TIF-GRPO** | **74.7** | **67.5** | **70.8** |

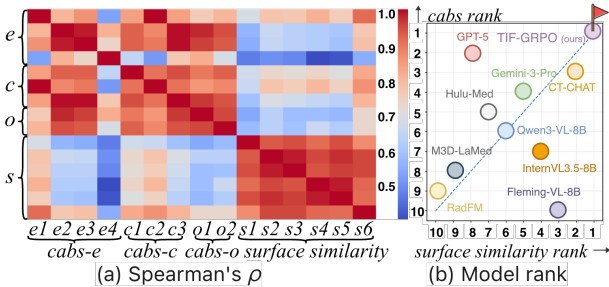

(a) Spearman's $\rho$      (b) Model rank

*Figure 5.* Evaluation Hallucination Analysis. *cabs-e, c, o* represent the Entity Core, Clinical Fidelity, Organ Coverage metrics in the CABS system, indicating the real clinical competence verified by radiologists. *s1-s6* represent the *surface similarity* metrics: BLEU, ROUGE, METEOR, RadGraph, RaTEScore, and BioBert Score.

metrics, which encompass Entity Core, Clinical Fidelity, and Organ Coverage. The correlation is computed over the rank vectors of the models on each metric, mitigating scale dependency. First, as shown in Fig 5 (a), the Spearman correlation heatmap reveals a clear block-diagonal structure. While metrics within each suite are highly correlated, the correlations between surface-similarity and CABS metrics are consistently weak. This indicates that these two suites measure fundamentally distinct capabilities: surface fluency is not a proxy for clinical factuality. Second, Fig 5 (b) plots each model's rank under the two evaluation paradigms. TIF-GRPO achieves the top rank in both, demonstrating aligned optimization. In contrast, models like Fleming-VL and InternVL3.5 occupy the bottom-right corner, exhibiting high surface-similarity ranks but low CABS (clinical) ranks, a hallmark of evaluation hallucination. Conversely, GPT-5 resides in the Top-left, with a modest surface-similarity score but a leading CABS rank, consistent with its known gains in clinical reasoning. Together, these analyses provide empirical evidence for the "*Evaluation Hallucination*" phenomenon: conventional surface-similarity metrics are systematically misaligned with true clinical capability.

### 4.5. Mechanistic Divergence in Medical RL

To empirically validate the "*Mechanistic Divergence*" hypothesis, we conduct a counterfactual ranking consistency analysis (Fig. 6): starting from a ground-truth (GT) report,

we generate a set of clinically plausible variants via controlled perturbations involving 0–5 abnormal entity modifications, preserving their *textual* clinical priority order (*Text-Rank*). Each evaluation metric then induces its own *Metric-Rank* over the same report pool. We quantify alignment fidelity via the **concordance ratio** $\phi = P/\binom{n}{2}$, the fraction of report pairs whose relative ordering is preserved between Text-Rank and Metric-Rank.

Results on over 2000 reports from CT-RATE Fig. 6 (a) and 1000 reports from AMOS-MM Fig. 6 (b) reveal a stark divergence: surface-similarity metrics (BLEU, ROUGE, METEOR) achieve only moderate $\phi$ ($\approx$0.65–0.75), indicating they frequently misorder clinically critical reports. Even structured semantic metrics like RadGraph and RaTE-Score show limited fidelity ($\phi < 0.8$). In contrast, **CABS-F1 achieves the highest concordance** ($\phi > 0.90$), demonstrating that its decomposition into verifiable clinical semantic units enables robust discrimination of clinical priority. This confirms that reward signals derived from surface metrics induce optimization trajectories decoupled from clinical truth, while CABS-based rewards align policy updates with genuine clinical semantics, directly mitigating collapse into

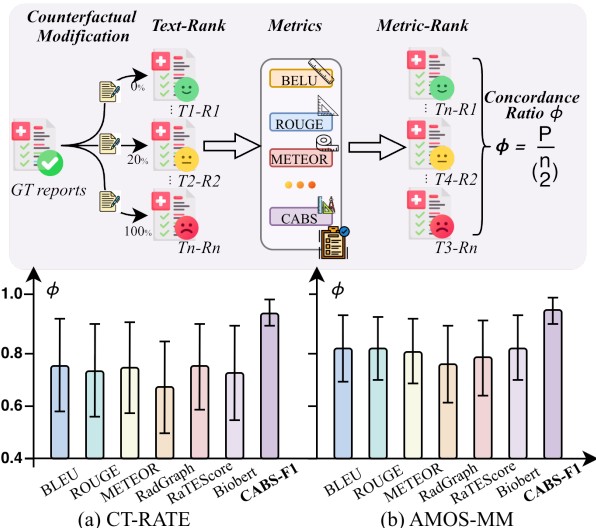

*Figure 6.* Mechanistic Divergence Analysis via counterfactual ranking consistency evaluation. We perturb GT reports to generate clinically plausible variants (0–5 abnormal entity modifications), whose Text-Rank reflects clinical priority. Concordance ratio $\phi = P/\binom{n}{2}$ measures pairwise rank agreement. CABS-F1 achieves the highest $\phi$, indicating superior clinical fidelity.

*Table 5.* Ablation Analysis of different reward signals and TIF-GRPO. $P \downarrow$ indicates that the model reports fewer abnormal cases. $N \downarrow$ indicates fewer normal cases.

| Training Strategy | Precision | Recall | F1 | Loc-Acc | Att-Acc |
|---|---|---|---|---|---|
| SFT | 34.59 | 12.85 | 18.73 | 67.99 | 52.22 |
| GRPO + ROUGE | 37.18 | 24.81 | 29.74 | 64.55 | 54.86 |
| GRPO + RadGraph | 35.39 | 19.73 | 25.32 | 57.29 | 55.22 |
| GRPO + Biobert | 37.82 | 23.17 | 28.7 | 68.35 | 57.92 |
| GRPO + LLM | 38.18 | 28.64 | 32.73 | 68.91 | 59.04 |
| TIF-GRPO ($\frac{\alpha}{\gamma} = \frac{0.4}{1.0}$) | $46.34_{P\downarrow}$ | $31.07_{P\downarrow}$ | 36.96 | **71.62** | 62.18 |
| TIF-GRPO ($\frac{\alpha}{\gamma} = \frac{1.0}{0.4}$) | $34.27_{N\downarrow}$ | $41.45_{N\downarrow}$ | 37.32 | 69.51 | 61.90 |
| TIF-GRPO ($\frac{\alpha}{\gamma} = \frac{1.0}{1.0}$) | 40.93 | 37.80 | **39.11** | 70.23 | **65.79** |

safe modes that neglect long-tail abnormalities.

### 4.6. TIF-GRPO Analysis

We construct rewards from surface similarity metrics to study how mechanistic divergence impacts GRPO training (Table 5). Consistent with our earlier finding that RadGraph poorly discriminates samples, it yields the weakest gains when used as the reward. Although one reward is LLM-based, fine-grained guidance grounded in clinical semantic units provides clear advantages. By adjusting the signal ratios in TIF-GRPO, we further observe that Running Cost and Control Effort reduce false positives and false negatives, achieving the best Precision and Recall, respectively; balanced weighting delivers the highest F1, underscoring their complementarity.

To test whether this divergence is inherent to surface-level similarity rewards rather than specific to GRPO, we evalu-

*Table 6.* Ablation Analysis of different RL algorithms.

| Training Strategy | Precision | Recall | F1 | Loc-Acc | Att-Acc |
|---|---|---|---|---|---|
| SFT | 34.59 | 12.85 | 18.73 | 67.99 | 52.22 |
| GRPO + ROUGE | 37.18 | 24.81 | 29.74 | 64.55 | 54.86 |
| GRPO + LLM | 38.18 | 28.64 | 32.73 | 68.91 | 59.04 |
| RLOO + ROUGE | 36.76 | 23.97 | 29.01 | 62.71 | 54.78 |
| RLOO + LLM | **41.24** | 30.71 | 35.20 | 67.04 | 57.93 |
| ReMAX + ROUGE | 32.93 | 25.78 | 28.92 | 58.22 | 52.89 |
| ReMAX + LLM | 38.33 | 27.94 | 31.71 | 66.57 | 57.56 |
| Reinforce++ + ROUGE | 36.72 | 29.26 | 32.56 | 65.33 | 58.12 |
| Reinforce++ + LLM | 35.89 | 27.53 | 31.16 | 68.34 | 60.11 |
| TIF-GRPO | 40.93 | **37.80** | **39.11** | 70.23 | **65.79** |

ate three additional RL methods (Table 6). They provide moderate improvements over SFT but consistently lag behind TIF-GRPO, indicating that mechanistic divergence is a broader issue in RL for medical long-form generation when surface similarity metrics serve as optimization signals.

## 5. Conclusion

We identify a critical misalignment in medical VLMs: proxy-based evaluation and RL rewards induce "*Evaluation Hallucination*" and "*Mechanistic Divergence*", steering models away from clinical factuality. To address this, we propose CABS, a structured system that decomposes reports into verifiable abnormality units for precise organ- and finding-level assessment. Utilizing CABS, we design a clinically grounded reward and TIF-GRPO, a trajectory-integral RL framework that stabilizes optimization through anatomy-aware integral feedback. Our method achieves SOTA on multiple 3D CT benchmarks, significantly improving diagnostic fidelity and establishing a new paradigm for fine-grained regulation of post-training in medical VLMs.

## Acknowledgements

This work has been supported in part by the NSFC (No. 62436007), the China Postdoctoral Science Foundation under Grant Number 2024M752794, the ZJNSF (No. LZ25F020004), the Key Research and Development Projects in Zhejiang Province (No. 2025C01128, 2025C02156), Ningbo Yongjiang Talent Introduction Programme (2023A-400-G).

## Impact Statement

This work aims to improve factuality and faithfulness in radiology report generation for medical vision-language models. Our approach may help reduce clinically misleading generations and support more reliable evaluation of medical AI. However, these models can still produce factual errors or miss critical findings, and thus any practical deployment should remain under strict clinical oversight, external validation, and appropriate safety and data-governance safeguards.

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

# Appendix

# A. Related Work

## A.1. Medical Vision-Language Models

Medical VLMs have evolved from cross-modal alignment to unified, reasoning-capable systems. BioMedGPT (Zhang et al., 2023) laid sequence-to-sequence foundations; LLaVA-Med (Li et al., 2023a) and HuatuoGPT-Vision (Chen et al., 2024b) enhanced alignment and data quality via curriculum learning and denoising; Med-Flamingo (Moor et al., 2023) enabled few-shot adaptation; HealthGPT (Lin et al., 2025) and Lingshu (Xu et al., 2025) addressed task interference and refined medical understanding; and MedGemma (Sellergren et al., 2025) achieved high-order reasoning with medically-tuned MedSigLIP encoders. Moreover, given the substantial heterogeneity across medical specialties in terms of evidence modalities, diagnostic workflows, and clinical reasoning paradigms, recent studies have increasingly focused on specialty-specific training and optimization. For example, works such as EyeCareGPT (Li et al., 2025b), HeartCareGPT (Xie et al., 2026), and OralGPT-Omni (Hao et al., 2025) explicitly tailor model capabilities to the unique characteristics of their respective clinical tasks. In the 3D domain, models progressed from slice-wise to native volumetric processing: RadFM (Wu et al., 2025a) established benchmarks; CT2Rep (Hamamci et al., 2024b) pioneered 3D CT report generation; Med3DInsight (Chen et al., 2024c) and M3D-LaMed (Bai et al., 2024) bridged 2D–3D gaps via plane-aware attention and spatial pooling; 3D-CT-GPT (Chen et al., 2024a) enabled direct diagnostic generation. Recent Med-2E3 (Shi et al., 2024), Hulu-Med (Jiang et al., 2025), Fleming-VL (Shu et al., 2025) and OmniCT (Lin et al., 2026) pursue unified perception for 2D/3D medical data.

## A.2. Medical VLMs Metrics and Evaluation

Accurately assessing the quality of generated medical reports remains a pivotal challenge. Traditional NLP metrics rely heavily on lexical overlap, leading to a severe "*Evaluation Hallucination*" where reports with high scores may contain critical clinical errors (Pino et al., 2020). To mitigate this, RadGraph (Jain et al., 2021) quantifies anatomical entities and pathology relations via structured graphs. Subsequent frameworks like RaTEScore (Zhao et al., 2024), ReEvalMed (Li et al., 2025a), and the Semantic Evaluation Framework (Ali et al., 2025) attempt to capture clinical consistency through semantic matching or LLM-assisted scoring.

However, these methods remain proxy signals that aim to align textual similarity but fail to capture the fine-grained coupling among anatomical ontology, location, and clinical evidence grounded in the image. In contrast, our CABS decomposes free-text reports into structured abnormality units explicitly aligned with clinical facts, establishing a measurable coordinate system that shifts the training objective from surface similarity to clinical fidelity.

## A.3. Reinforcement Learning in Medical VLMs

Reinforcement Learning for LLM has evolved from foundational algorithms like PPO (Schulman et al., 2017) to efficient, critic-free methods like ReMax (Li et al., 2023b), RLOO (Ahmadian et al., 2024), and GRPO (Shao et al., 2024), enabled group-relative optimization without value models. To improve stability, Reinforce++ (Hu et al., 2025) introduced global advantage normalization, and GSPO (Zheng et al., 2025) optimized sequence-level importance ratios to stabilize training for Mixture-of-Expert architectures. Algorithms like DAPO (Yu et al., 2025)and DrGRPO (Liu et al., 2025a) addressed optimization biases and dynamic sampling issues, while PRIME (Cui et al., 2025a) provided dense implicit process rewards to mitigate reward hacking. Furthermore, KL_Cov and Clip_Cov(Cui et al., 2025b) have been proposed to mitigate policy entropy collapse and performance saturation during scaling.

In the medical domain, RL is increasingly applied to enhance diagnostic logic and transparency. Dr-LLaVA (Sun et al., 2024) pioneered symbolic clinical grounding to align models with established diagnostic pathways. Inspired by the success of reasoning models, MedVLM-R1 (Pan et al., 2025) and Med-R1 (Lai et al., 2025) demonstrated that models can exhibit emergent Chain-of-Thought capabilities across modalities without expert-annotated traces. Specialized applications include PathVLM-R1 (Wu et al., 2025b) and Patho-R1 (Zhang et al., 2025) , which integrate process rewards for pathology reasoning, and Skin-R1 (Liu et al., 2025b) , which incorporates hierarchical disease structures. In radiology, ChestX-Reasoner (Fan et al., 2025) and MedReason-R1 (Li et al., 2025c) focus on step-by-step verification and local zoom mechanisms to mimic clinical reading, while QoQ-Med (Dai et al., 2025) employs domain-aware rewards to balance modality heterogeneity. To overcome data scarcity, MedGR2 (Zhi et al., 2025) builds self-evolving generative reward loops, and MedVLThinker (Huang et al., 2025) establishes high-performance baselines for reasoning-centric transfer learning. However, these efforts typically apply general RL policy with simple or clinical-sparse signal as reward.

# B. Experimental Details

## B.1. Dataset and preprocessing details

We evaluate our method on two public multimodal CT datasets: CT-RATE (Hamamci et al., 2024c) and AMOS-MM (Ji et al., 2022). CT-RATE (CT with Radiology Reports And Text Evaluation) is a public chest CT multimodal dataset that pairs 3D chest CT volumes with their radiology reports and labels for 18 abnormalities. It contains more than 25k non-contrast scans from about 21k patients and is expanded to roughly 50k volumes via different reconstructions. The dataset was originally released with CT-CLIP and has since been widely used for 3D medical vision–language pretraining and evaluation. AMOS-MM is the multimodal extension of the AMOS benchmark, built on large-scale abdominal CT. It provides paired images and radiology reports (including both Findings and Impression), together with structured labels and question–answer annotations, and serves as a benchmark for vision-language modeling and evaluation on abdominal and torso CT. During abnormal entity extraction in CABS, we employ GPT-5 (OpenAI, 2025) to generate structured candidate entities from the extracted content. For TIF reward computation, we separately utilize Qwen3-30B-A3B (Yang et al., 2025) to evaluate whether the identified abnormal entities are correctly matched.

Based on the two datasets mentioned above, we further construct multiple-choice VQA subsets for both CT-RATE and AMOS-MM, denoted as **CT-RATE-MCQ** and **AMOS-MM-MCQ**, respectively (collectively referred to as our MCQ datasets). The construction procedure consists of two stages. First, we use a carefully designed prompt in Appendix G to drive GPT-5 (OpenAI, 2025) that extracts structured abnormality entities from each ground-truth radiology report (Findings or Impression). For every report, the model outputs entities with the fields *name*, *evidence*, *location*, *attributes*, *certainty*, and *organ*. The extraction strictly follows an evidence-driven principle: we only keep abnormalities that are explicitly stated in the report, discard negated findings, normal anatomy, and purely technical descriptions, and merge repeated descriptions of the same abnormality into a single entity.

In the second stage, each structured abnormality entity is paired with a *negative_name* that is guaranteed to be absent in that specific case (sampled from abnormality names in the entire dataset but excluded from the current study), and fed into another VQA-construction prompt in Appendix G. This prompt guides a large model to automatically generate 2–4 English multiple-choice questions that can be answered from the CT images alone. For each entity, we always create one positive existence question (*existence_positive*, correct answer *Yes*) asking whether the abnormality is present, and one negative existence question (*existence_negative*, correct answer *No*) about the *negative_name*. If *location* is non-empty, we additionally generate a four-choice location question (*location*); if *attributes* is non-empty, we generate a four-choice attribute question (*attribute*). Existence questions only use two options ("Yes" / "No"), whereas location and attribute questions have four options, with exactly one option semantically consistent with the structured label, and the others being plausible but incorrect distractors. Throughout the construction of CT-RATE-MCQ and AMOS-MM-MCQ, both questions and options are strictly constrained not to quote or paraphrase the report text and must be phrased in an image-centric manner (e.g., "On this chest CT . . .", "In this examination . . ."), ensuring that VQA models are evaluated on genuine image-based reasoning rather than exploiting report priors.

## B.2. Training Parameters

To facilitate reproducibility, we provide a comprehensive summary of the training configurations used throughout all experimental stages. We utilize LLaMA-Factory (Zheng et al., 2024) for supervised fine-tuning (SFT) and the Verl framework (Sheng et al., 2024) for TIF-GRPO optimization.

The reported configurations cover both optimization and system-level settings, including learning rates, scheduler types, batch sizes, sequence lengths, gradient accumulation steps, precision formats, rollout configurations, and distributed training strategies. Detailed SFT hyperparameters are presented in Table 9, while the TIF-GRPO settings are summarized in Table 10.

## B.3. Method Symbol Description

For clarity and ease of reference, we summarize the key mathematical symbols and notations used throughout the paper in Table 8. The table includes the definitions of variables, optimization objectives, intermediate representations, and training-related quantities appearing in both the methodology and experimental sections. We maintain consistent notation across all stages of the proposed framework to avoid ambiguity and improve readability.

| Dimension | Concept | Score Definition |
|---|---|---|
| **Clinical Correctness** | Whether the decomposed abnormality entities are medically consistent with the original radiology report, including correctness of abnormality existence, organ / site / laterality, and key attributes (e.g., size, severity, density) | Scores (1–5):
**5**: completely correct.
**4**: minor issues, overall correct.
**3**: mix of correct and incorrect.
**2**: many errors.
**1**: largely wrong. |
| **Evidence Traceability** | Whether each abnormality entity can be directly supported by explicit textual evidence from the original report, and whether the evidence clearly substantiates the abnormality and its key descriptions. | Scores (1–5):
**5**: clear, direct span.
**4**: relevant span.
**3**: needs inference.
**2**: weak or ambiguous span.
**1**: missing or mismatched span. |
| **Coverage Integrity** | Whether the major abnormalities described in the original report are comprehensively decomposed, with no important findings omitted. | Scores (1–5):
**5**: no important misses.
**4**: only minor misses.
**3**: some important misses.
**2**: many misses.
**1**: key abnormalities missing. |
| **Clinical Decomposition Usability** | Whether the overall decomposition structure is clinically and structurally usable, considering over-/under-splitting, alignment with clinical reasoning, and suitability for downstream tasks. | Scores (1–5):
**5**: directly usable.
**4**: usable with small fixes.
**3**: usable with moderate edits.
**2**: hard to use in practice.
**1**: essentially unusable. |

*Table 7.* Evaluation dimensions and scoring criteria for abnormality decomposition on CT reports.

## C. Metric Details

We introduce the detailed computation process of evaluation metrics here, including 8 metrics in the CABS system and 6 conventional surface-level similarity metrics. In CABS, the ground-truth report is first decomposed into a set of reference abnormality entities $\mathcal{F}^{gt} = \{f_1, \ldots, f_K\}$, and the model prediction is mapped into a corresponding set of predicted entities $\mathcal{F}^{pred}$. We then perform one-to-one entity matching under the evidence-based abnormality alignment protocol described in the prompt templates. Each ground-truth entity is assigned three binary indicators: *hit*, *location_match*, and *attribute_match*. Importantly, entity detection and fine-grained fidelity are evaluated separately.

**Entity Core.** This group measures whether the model detects the correct abnormality entities, regardless of whether the associated location or attribute description is correct. In other words, an entity is counted as a true positive as long as the abnormality itself is correctly identified (*hit = true*); errors in location or attribute do *not* affect the hit decision at this stage. Let

$$TP_e = \sum_{f \in \mathcal{F}^{gt}} \mathbf{1}[\text{hit}(f) = 1], \quad FN_e = \sum_{f \in \mathcal{F}^{gt}} \mathbf{1}[\text{hit}(f) = 0],$$

and let $FP_e$ denote the number of unmatched false-positive abnormality entities extracted from $\mathcal{F}^{pred}$. Then:

$$\text{Precision} = \frac{TP_e}{TP_e + FP_e + \varepsilon}, \quad \text{Recall} = \frac{TP_e}{TP_e + FN_e + \varepsilon},$$

$$\text{F1} = \frac{2 \cdot \text{Precision} \cdot \text{Recall}}{\text{Precision} + \text{Recall} + \varepsilon}.$$

Therefore, Entity Core strictly answers the question: *did the model detect the right abnormality entity, and did it avoid inventing false ones?*

*Table 8.* Nomenclature and Concept Definitions

| Symbol | Concept | Symbol | Concept |
|---|---|---|---|
| $\mathcal{V}$ | The set of 3D CT volumes | $\mathcal{F}$ | Set of discrete, verifiable clinical semantic units |
| $V$ | A specific input 3D CT volume | $f_i$ | The $i$-th clinical semantic unit (abnormality) |
| $Z, H, W$ | Dimensions of CT (Depth, Height, Width) | $\mathcal{S}$ | Predefined clinical ontology space |
| $q$ | Clinical task prompt | $o$ | Target organ within the ontology |
| $\theta$ | Parameters of Large VLM | $d$ | Abnormal entity within the ontology |
| $y$ | Generated clinically faithful response | $\mathcal{A}$ | Set of pathological attributes |
| $w_t$ | The $t$-th token in the response sequence | $l$ | Anatomical location precision |
| $\pi_\theta$ | Conditional policy of the model | $c$ | Diagnostic certainty |
| $y_{gt}$ | Ground-truth reports | $e$ | Textual evidence extracted from report |
| $\mathcal{L}_{SFT}$ | Supervised Fine-Tuning (SFT) loss | $\Phi$ | Mapping function from response to clinical facts |
| $\mathcal{D}$ | Training dataset | $y_{gen}$ | Generated response |
| $G$ | Group size for GRPO sampling | $y_{ref}$ | Reference report |
| $\pi_{\theta_{old}}$ | Old policy model prior to update | $r_i$ | Individual hit reward for unit $f_i$ |
| $R_G$ | Set of rewards for the sampled group | $\text{hit}_i$ | Binary indicator for relevant concept hit |
| $A_i^{GRPO}$ | Normalized group advantage | $l_i$ | Indicator for correct location identification |
| $\mu_G$ | Mean of the reward set $R_G$ | $d_i$ | Indicator for correct entity identification |
| $\sigma_G$ | Standard deviation of the reward set $R_G$ | $R_{CABS}$ | Global instantaneous reward based on CABS |
| $\epsilon$ | Small constant to avoid division by zero | $R_{TIF}$ | Trajectory-Integral Reward |
| $\mathcal{J}(\theta)$ | Optimization objective function | $\alpha$ | Scalar hyperparameter (Running Cost term) |
| $\lambda_i$ | Importance ratio $(\pi_\theta/\pi_\theta^{old})$ | $FP$ | Number of false positives |
| $D_{KL}$ | Kullback–Leibler divergence | $M$ | Number of predicted abnormal entities |
| $\pi_{ref}$ | Reference model | $\mathbf{1}[\cdot]$ | Sign/Indicator function |
| $\beta$ | Weight of KL loss / scalar in $R_{TIF}$ | $A_i^{TIF}$ | Trajectory-Regulated Advantage |
| - | - | $\mathcal{J}^{TIF}(\theta)$ | Optimization objective for TIF-GRPO |

*Table 9.* Main parameters of Llama-Factory Training.

| Parameter | Value |
|---|---|
| Finetuning Type | Full |
| Precision | bf16 |
| Learning Rate | 2e-5 |
| Optim | Adamw |
| LR Scheduler Type | Cosine |
| Warmup Ratio | 0.1 |
| Per Device Train Batch Size | 8 |
| Gradient Accumulation Steps | 1 |
| Weight Decay | 0.0 |
| Num Train Epochs | 1 |

*Table 10.* Main parameters of the Verl Training.

| Parameter | Value |
|---|---|
| Advantage Estimator | grpo |
| Train Batch Size | 128 |
| Max Response Length | 2048 |
| Actor Optim LR | 2e-6 |
| PPO Mini Batch Size | 64 |
| PPO Micro Batch Size Per GPU | 4 |
| Use kl Loss | True |
| Entropy Coeff | 0.0 |
| GPU Memory Utilization | 0.65 |
| Total Epochs | 1 |

**Clinical Fidelity.** This group evaluates the quality of the description *conditioned on successful detection*. We only consider the subset of hit entities,

$$\mathcal{F}^{hit} = \{f \in \mathcal{F}^{gt} \mid \text{hit}(f) = 1\}.$$

Among these correctly detected abnormalities, we compute:

$$\text{Location Accuracy} = \frac{\sum_{f \in \mathcal{F}^{hit}} \mathbf{1}[\text{location\_match}(f) = 1]}{|\mathcal{F}^{hit}| + \varepsilon},$$

$$\text{Attribute Accuracy} = \frac{\sum_{f \in \mathcal{F}^{hit}} \mathbf{1}[\text{attribute\_match}(f) = 1]}{|\mathcal{F}^{hit}| + \varepsilon},$$

$$\text{Fully-Consistent Accuracy} = \frac{\sum_{f \in \mathcal{F}^{hit}} \mathbf{1}[\text{location\_match}(f) = 1 \wedge \text{attribute\_match}(f) = 1]}{|\mathcal{F}^{hit}| + \varepsilon}.$$

Hence, Clinical Fidelity does not ask whether the abnormality was detected at all; rather, it asks whether, *after a correct hit*, the model also provides the correct anatomical localization, the correct attribute description, and the fully consistent combination of both.

**Organ Coverage.** This group measures coverage at the organ level by first aggregating over abnormality entities within each organ and then averaging across organs. Let $\mathcal{O}^{gt}$ be the set of organs appearing in the ground-truth report, and let

$$\mathcal{F}_o^{gt} = \{f \in \mathcal{F}^{gt} \mid \mathrm{organ}(f) = o\}$$

denote the set of ground-truth abnormality entities belonging to organ $o$.

For Or-Rate, we only consider whether each abnormality entity is hit, regardless of whether its location or attribute is correct. We first compute the within-organ hit rate:

$$\mathrm{Or}(o) = \frac{\sum_{f \in \mathcal{F}_o^{gt}} \mathbf{1}[\mathrm{hit}(f) = 1]}{|\mathcal{F}_o^{gt}| + \varepsilon}.$$

We then average these organ-specific scores across all organs:

$$\mathrm{Or\text{-}Rate} = \frac{\sum_{o \in \mathcal{O}^{gt}} \mathrm{Or}(o)}{|\mathcal{O}^{gt}| + \varepsilon}.$$

For FMOr-Rate, the aggregation procedure is identical, except that an entity is counted as correct only when it is hit and simultaneously has correct location and correct attribute. We therefore define the within-organ fully matched rate as

$$\mathrm{FMOr}(o) = \frac{\sum_{f \in \mathcal{F}_o^{gt}} \mathbf{1}[\mathrm{hit}(f) = 1 \wedge \mathrm{location\_match}(f) = 1 \wedge \mathrm{attribute\_match}(f) = 1]}{|\mathcal{F}_o^{gt}| + \varepsilon},$$

and then compute the macro-average across organs:

$$\mathrm{FMOr\text{-}Rate} = \frac{\sum_{o \in \mathcal{O}^{gt}} \mathrm{FMOr}(o)}{|\mathcal{O}^{gt}| + \varepsilon}.$$

Thus, Or-Rate measures whether the model covers the abnormality entities of each organ at all, while FMOr-Rate is stricter and additionally requires that the matched entities preserve both correct location and correct attribute information.

**Conventional surface-level similarity.** For completeness, we also report BLEU, ROUGE, METEOR, RadGraph, RaTEScore, and BioBERT Score as conventional proxy metrics for comparison with the CABS-based clinical metrics above.

## D. Clinical Radiologist Evaluation

*Table 11.* Expert Fine-Grained Evaluation Scores

| Evaluation Demension | junior1 | junior2 | junior3 | senior1 | senior2 | senior3 |
|---|---|---|---|---|---|---|
| Clinical Correctness | 4.94 | 4.75 | 4.83 | 4.98 | 4.87 | 4.92 |
| Evidence Traceability | 4.99 | 4.93 | 4.95 | 4.98 | 4.92 | 4.94 |
| Coverage Integrity | 4.93 | 4.78 | 4.84 | 4.80 | 4.82 | 4.79 |
| Clinical Decomposition Usability | 4.86 | 4.74 | 4.74 | 4.87 | 4.79 | 4.78 |

Table 11 presents expert evaluation results for CABS. The consistently high scores across all dimensions (4.73–5.00) demonstrate that CABS produces high-quality abnormality entity units that are both clinically reliable and practically usable. Notably, Evidence Traceability is near-perfect (4.92–4.99), showing that CABS almost always grounds each extracted entity in clear, explicit evidence from the original report—directly validating its traceability-oriented design. Clinical Correctness is likewise strong (4.75–4.98), indicating that the decomposed entities closely match the medical content of the reports. Meanwhile, the high Coverage and Usability scores suggest CABS captures the vast majority of relevant findings and outputs a decomposition structure that is readily applicable to downstream tasks with minimal refinement.

## E. More Experiments

In Section 4.6, we showed that running cost and control effort constitute essential mechanisms for clinically usable reporting: running cost encourages the model to identify abnormalities, whereas control effort suppresses false-positive reporting.

CT-RATE-Report: Reward

CT-RATE-Report: Entropy

CT-RATE-Report: Response

*Figure 7.* Impact of Running Cost weights and Control Effort weights on the Training Dynamics of TIF-GRPO.

Beyond the quantitative results, we further provide training curves in Table 5 to illustrate how these terms shape training behaviors: **(1) Reward:** TiF-GRPO converges stably whether the weights of running cost and control effort are balanced or dominated by one side. We observe no reward hacking (e.g., staying completely silent or spamming abnormality reports to "game" the reward), demonstrating the stability and convergence efficiency of TiF-GRPO. **(2) Entropy:** When running cost dominates, TiF-GRPO exhibits stronger exploration and thus higher entropy; when control effort dominates, the policy becomes more conservative, avoiding excessive probing that leads to false positives. With balanced weights, the model achieves an effective trade-off between abnormality detection and error avoidance. **(3) Response:** Under a dominant running cost, the model tends to produce longer responses, increasing the chance of covering abnormal cues and hitting true abnormalities. In contrast, when control effort dominates, the model prefers reporting only high-confidence abnormalities, resulting in shorter and more restrained responses.

# F. Cases Analysis

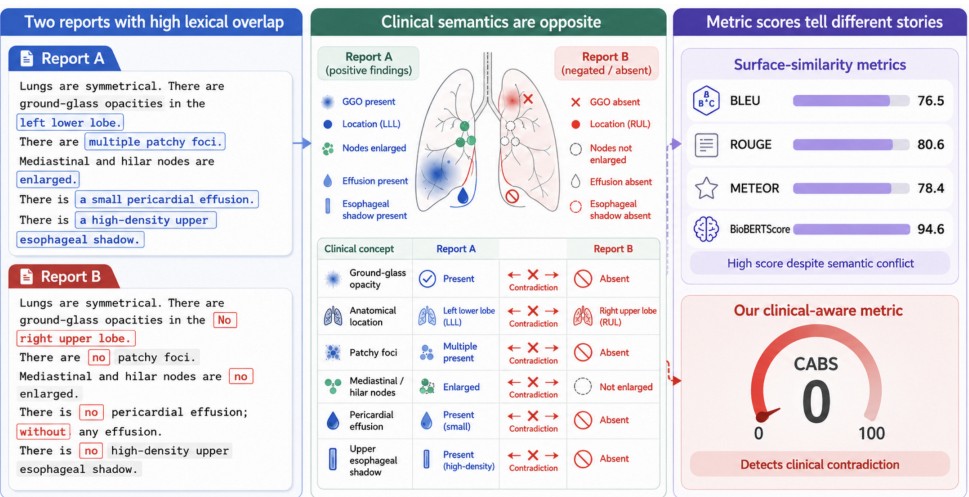

*Figure 8.* Case study on surface similarity metrics and CABS.

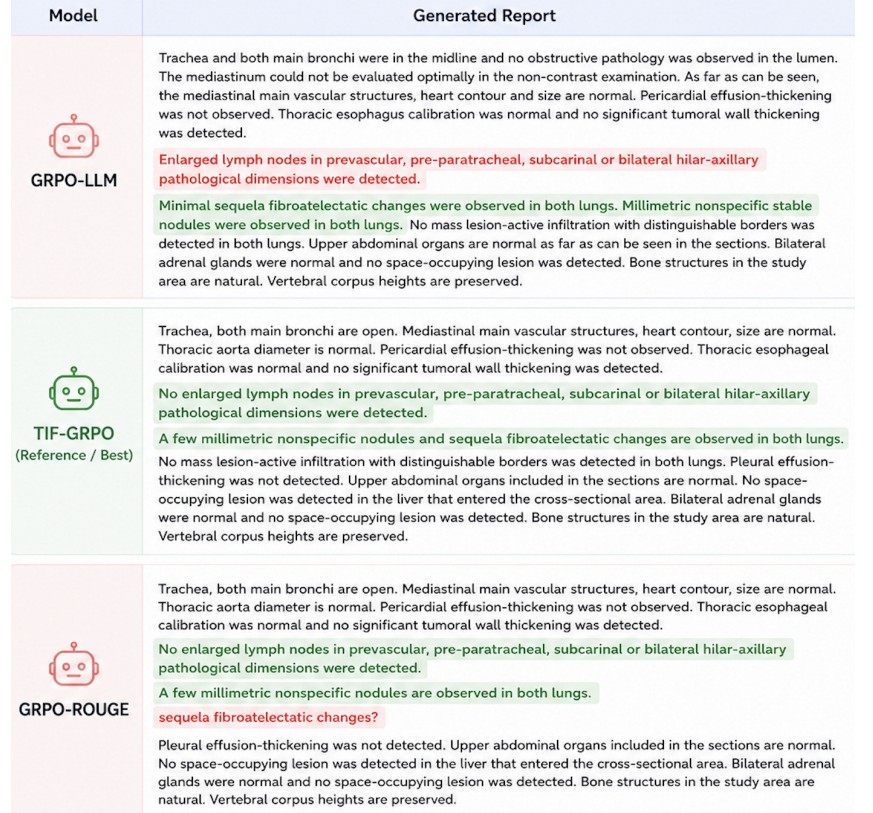

*Figure 9.* Case study on TIF-GRPO, GRPO-ROUGE and GRPO-LLM.

## G. Prompt templates.

---

**Prompt for Constructing CT-RATE-MCQ and AMOS-MM-MCQ**

**Task.** Generate multiple-choice VQA questions in English for chest CT examinations.

**Core premise.** The answering model only sees CT images. The structured abnormal labels extracted from the report are available only to the annotator as ground truth. Therefore, every question must be answerable from visible CT evidence alone and must not quote, paraphrase, or reveal report content or label conclusions.

**Inputs.**

1. One target abnormality object from the case, with fields such as `name`, `evidence`, `location`, `attributes`, and `organ`.
2. One `negative_name`, which is guaranteed not to appear in this case and is used to construct the negative existence question.

**Question types and counts.** The prompt must generate between 2 and 4 questions in English only.

1. `existence_positive`: must ask whether the current abnormality is present on the chest CT. Example phrasings include "On this chest CT, is there a 'soft tissue mass' abnormality?" and "On this chest CT, is a 'soft tissue mass' visible?" The report must not be mentioned. Options must be exactly two choices, `Yes/No` or `No/Yes`. Since the abnormality is present, the correct answer must be `Yes`.
2. `existence_negative`: must ask whether `negative_name` is present on the chest CT. The question must remain image-based, e.g., "On this chest CT, is there a '<negative_name>' abnormality?" The report must not be mentioned. Options must again be exactly two choices, and the correct answer must be `No`.
3. `location` (optional): generate this question only if `abnormality.location` is non-empty. It should ask where the abnormality is mainly located on the CT images, e.g., "On this chest CT, where is the 'soft tissue mass' mainly located?" or "On this image, in which location is the 'soft tissue mass' visible?" Options must be exactly four anatomical location phrases. At least one option must be semantically equivalent to the ground-truth location, and the options may span different regions to maximize diversity.
4. `attribute` (optional): generate this question only if `abnormality.attributes` is non-empty and meaningful. It should ask how the abnormality appears on the image, e.g., "On this chest CT, which of the following statements best describe the appearance of this 'soft tissue mass'?" or "On this CT image, how does this 'soft tissue mass' appear?" Options must be exactly four imaging attribute phrases. At least one option must reflect the semantics of the ground-truth attributes, while the other options should be plausible but incorrect for this case.

**Summary constraints.**

- There must be exactly one `existence_positive` question.
- There must be exactly one `existence_negative` question.
- If `abnormality.location` is non-empty, there must be one `location` question; otherwise none.
- If `abnormality.attributes` is non-empty, there must be one `attribute` question; otherwise none.
- The total number of items must therefore be between 2 and 4.

**Wording rules.**

- All questions and options must be written in English.
- Every question must be answerable from CT images alone.
- The final VQA model will not see the report or the structured labels.
- The question text must not mention "report", "findings", "impression", or any report field, nor expressions such as "in this report", "according to the report", or "described in the report".
- Questions should instead be phrased in an image-centric way, such as "On this chest CT . . . " or "In this examination . . . ".
- The options should also be phrased as visual findings.

**Case placeholders.** The prompt receives the target abnormality as {`abnormality_json`} and the absent disease name as {`negative_name`}.

**Example output.**

```
"items": [

    "type": "existence_positive / existence_negative / location / attribute",
    "question": "English question text (image-based, no mention of the report)",
    "options": ["A. ...", "B. ...", "C. ...", "D. ..."],
    "answer": "A" / "B" / "C" / "D"

]
```

**Output notes.**

---

- For `existence_positive` and `existence_negative`, options are restricted to the binary yes/no setting.
- For `location` and `attribute`, exactly four options must be provided.
- The `answer` field must contain only option letters, such as A, B, C, or D.
- No explanations or extra keys may appear outside the JSON structure.

## Prompt for Constructing Abnormal Units

**Task.** Extract abnormality entities from a single ground-truth CT report and output them as a structured JSON object with high precision.

**Input report.** The ground-truth Findings or Impression text is provided as the placeholder {`report`}.

**Step 1: abnormality definition.** An abnormality entity is any medical concept explicitly described as abnormal, pathological, or non-normal, including diseases (e.g., pneumonia, fatty liver), abnormal imaging findings (e.g., ground-glass opacity, nodule), and structural abnormalities (e.g., effusion, calcification, enlargement, stenosis).

**Do not extract the following.**

1. Explicitly negated findings, such as "no", "not seen", or "negative for".
2. Normal anatomical structures or normal imaging findings.
3. Content that only describes scanning conditions, image quality, or technical limitations.
4. Recommendations, follow-up suggestions, or examination plans without an explicit abnormality statement.

**Step 2: extraction rules.**

- **Evidence-only extraction:** extract only abnormalities explicitly stated in the report. Do not infer, assume, or hallucinate.
- **Preserve uncertainty:** if the report says "possible", "cannot exclude", or "consider", this uncertainty must be preserved rather than upgraded to a definite finding.
- **Entity merging:** if multiple sentences clearly refer to the same abnormality, merge them into one entity.
- **Single-entity principle:** if one statement describes a unified abnormality concept, such as "multiple small nodules in both lungs", treat it as one entity unless the report explicitly distinguishes separate lesions.
- **Negation exclusion:** any abnormality that is explicitly negated or ruled out must be omitted.

**Step 3: required fields for each abnormality entity.**

- `name`: the standardized abnormality name only, without anatomical location, size, severity, modifiers, or attributes. Example values include `nodule`, `ground-glass opacity`, and `fatty liver`.
- `evidence`: a verbatim text span copied directly from the report, concise but sufficient to support the abnormality.
- `location`: the explicitly stated anatomical location; if absent, use an empty string.
- `attributes`: imaging or pathological attributes such as size, number, extent, density, morphology, severity, or temporal change; if absent, use an empty string.
- `certainty`: must be one of `definite` or `possible`, based strictly on the report wording.
- `organ`: the normalized organ label, restricted to `trachea`, `heart`, `lung`, `esophagus`, `vessel`, `spine`, `liver`, `pancreas`, `spleen`, `stomach`, `bowel`, `kidney`, or `other`.

**Step 4: report-level field.** In addition to the abnormality list, the output must include `report_has_abnormality`, which is `true` if at least one abnormality entity is extracted and `false` otherwise.

**Final constraints.** Do not extract diagnostic interpretations, speculative explanations, or parenthetical inferences as independent abnormalities. Repeated descriptions of the same lesion should remain merged. Only one valid JSON object may be returned, with no explanations or additional text.

**Example output.**

```
    "abnormalities": [

        "name": "ground-glass opacity",
        "evidence": "Patchy ground-glass opacities are seen in the bilateral lower lungs with ill-defined margins",
        "location": "bilateral lower lungs",
        "attributes": "patchy distribution with ill-defined margins",
        "certainty": "definite",
        "organ": "lung"
      ,

        "name": "fatty liver",
        "evidence": "Diffuse decreased attenuation of the liver, consider fatty liver",
        "location": "liver",
        "attributes": "diffusely decreased attenuation",
        "certainty": "possible",
        "organ": "liver"
      ,

        "name": "pleural effusion",
        "evidence": "There is evidence of pleural effusion",
        "location": "",
        "attributes": "",
        "certainty": "definite",
        "organ": "lung"

    ],
    "report_has_abnormality": true
```

## Prompt for Reward Structured Unit Mapping

**Task.** Compare a ground-truth CT report and a model-predicted CT report in terms of abnormality detection and description consistency, and output one JSON object only.
**Inputs.**

- Ground-truth abnormalities of the Findings or Impression section, already extracted as structured JSON: {gt}.
- Model-predicted Findings or Impression text: {pred}.

**Step 1: abnormality extraction from the prediction.**

- Use the provided ground-truth JSON directly as the reference abnormality set.
- Do not modify, merge, split, infer, or add any new ground-truth entities.
- Extract abnormality entities from the model-predicted report under the same definition rules.
- Exclude explicitly negated findings, normal findings, examination-condition descriptions, and interpretive or etiological impressions that are not tied to explicit imaging abnormalities.
- The name field of each extracted predicted entity must contain only the standardized abnormality name, without location, severity, modifiers, or attributes.

**Step 2: abnormality matching.** Match each ground-truth abnormality entity to at most one predicted abnormality entity based primarily on abnormality semantics. Medically equivalent or highly similar expressions can be matched, but no broadening, generalization, or inference beyond explicit textual evidence is allowed. If a matching abnormality is present, set hit = true; otherwise, set hit = false. Any predicted abnormality that matches no ground-truth entity must be listed as a false positive.
**Step 3: consistency judgment rules.**

- hit: true if a medically equivalent abnormality is explicitly present in the predicted report; false otherwise.
- location_match: true if the predicted anatomical location is medically consistent with the ground-truth location, or if neither side explicitly contains location information. It must be false when hit = false.
- attribute_match: true if the predicted imaging or pathological attributes are medically consistent with the ground-truth attributes, or if neither side explicitly contains attribute information. It must be false when hit = false.
- All judgments must remain evidence-based; missing locations or attributes must not be inferred.

**Step 4: result aggregation.**

- Report the matching and consistency results for every ground-truth abnormality entity.
- Report all unmatched false-positive abnormality entities found in the predicted report.

**Additional strict constraints.** Do not infer, assume, hallucinate, or strengthen uncertain expressions. Do not add abnormalities based on clinical plausibility or typical associations. Only explicit, text-supported abnormalities may be matched. Negated or ruled-out findings must not be extracted.

**Example output.**

```
"abnormalities": [

    "name": "<ground-truth abnormality name>",
    "hit": true / false,
    "location_match": true / false,
    "attribute_match": true / false

],
"false_positive": [

    "name": "<false positive abnormality name>"

]
```

