# OpenReview forum: "Regulating Anatomy-Aware Rewards via Trajectory-Integral Feedback for Volumetric Computed Tomography Analysis"
_ICML.cc/2026/Conference — ICML 2026 regular_

### Official Review · Reviewer_MZDZ · 2026-03-10

**Soundness:** 2
**Presentation:** 1
**Significance:** 4
**Originality:** 3
**Overall Recommendation:** 4
**Confidence:** 4

**Summary:**

In this paper, the author studied an important question that indicates evaluation hallucinations in 3D CT analysis, which is very important for the community. The paper also proposed a method called TIF-GRPO for training better VLM in CT analysis. However, the validation of the proposed evaluation metrics remains insufficient.

**Compliance With Llm Reviewing Policy:**

Affirmed.

**Final Justification:**

The paper discussed the issue of evaluation hallucination in 3D CT analysis. And the authors propose CABS to address this problem. However, in the previous version, the authors used the acceptance ratio to validate CABS's effectiveness and did not provide examples to illustrate evaluation hallucination.

In the rebuttal, the authors present a rank comparison with human experts and show that the proposed CABS is best aligned with human expert judgment, thereby demonstrating its effectiveness. However, I still believe that this comparison could be evaluated on a wider range of models.

Thus, I raise my score to 4.

**Key Questions For Authors:**

- The evaluation hallucination issue is interesting and important. However, the current evidence is not yet fully convincing, since the claim is primarily validated using the proposed CABS framework itself (e.g., Figure 5). This creates a potential risk of circular validation. Could the authors provide a more independent validation to prove that surface similarity metrics suffer from evaluation hallucination?

- In addition, the paper does not provide examples of hallucinations on surface similarity metrics. Could the authors include some specific examples to demonstrate how surface-similarity metrics can lead to evaluation hallucination?

- Since CABS is the paper's core contribution, the metric's definition and calculation should be more clearly described. At present, it is difficult to understand how the metrics are computed. Could the authors provide a clearer description of the CABS calculation pipeline?

**Limitations:**

- The CABS metrics are the central idea of this paper. However, this paper has less details on how to calculate these metrics, which is hard to follow.
- The current validation relies on experts evaluating the evaluation results (acceptance ratio), which is not fully convincing. And it's also hard to understand what this high acceptance ratio means. It could be better to have experts evaluate model performance directly, instead of evaluating the evaluation results. And then compare their rankings with those produced by CABS and surface-similarity metrics, similar to Figure 5.

**Strengths And Weaknesses:**

- **Soundness**: The problem studied in this paper is very important. However, the validation of the effectiveness of CABS is not sufficient.
- **Presentation**: The writing is poor. Although CABS is the core contribution of the paper, the metrics themselves and their validation are not described clearly, which makes the nice idea hard to follow.
- **Significance**: The problem studied in this paper is very important for the community. Many surface-similarity measurements can lead to reward hacking. By studying evaluation metrics, this work is valuable for future research.
- **Originality**:  The idea is novel and provides valuable insights into VLM evaluation.

---

> ### Author Rebuttal · Authors · 2026-03-31
>
> We thank the reviewers for their careful and constructive feedback. We present our detailed responses below.
>
> **(Q1) Is the evidence for CABS and evaluation hallucination circular, and can the validation be made independent of CABS itself?**
>
> (A1) We thank the reviewer for raising concerns about the sufficiency of CABS validation and potential evaluation circularity. Figure 6 provides a CABS-independent validation of evaluation hallucination in surface-similarity metrics, by assessing their discriminative ability on carefully constructed, ordered perturbation samples. Details of this mechanism are provided in `Reviewer WzFG (A6)`.
>
> For details on the expert acceptance rate of the CABS extraction process, please refer to `Reviewer NqCX (A3)` due to space limitations.
>
> Following the reviewer’s suggestion, we further introduce two independent evaluation protocols on generated reports:
> (1) four radiologists (2 junior + 2 senior) perform blind ranking on 200 samples, yielding results fully consistent with CABS;
> (2) based on CT-RATE classification labels, we compute label-vector distances using a pretrained classification model, producing highly consistent rankings.
>
> Across all evaluation systems, TIF-GRPO consistently ranks first:
>
> |Model|CABS|Surface Similarity|Expert|RadBert|
> |-|-|-|-|-|
> |TIF-GRPO|1|1|1|1|
> |GPT-5|2|4|2|3|
> |CT-CHAT|3|2|3|2|
> |Hulu-Med|4|3|4|4|
> |M3D-LaMed|5|5|5|5|
> |RadFM|6|6|6|6|
>
> These evaluations are mutually independent yet lead to consistent conclusions, indicating that CABS captures image-aligned clinical signals rather than introducing circular reasoning.
>
> **(Q2) Could the authors include some specific examples to demonstrate how surface-similarity metrics can lead to evaluation hallucination?**
>
> Below is a representative example:
>
> * Report A: Lungs are symmetrical. Ground-glass opacities are seen in the left lower lobe with multiple patchy foci; enlarged mediastinal and hilar nodes are seen, with a small pericardial effusion and a high-density upper esophageal shadow.
> * Report B: Lungs are symmetrical. ~~No ground-glass opacities~~[x] are seen in the ~~right upper lobe~~[x] with ~~no patchy foci~~[x]; ~~no enlarged mediastinal and hilar nodes~~[x] are seen, ~~without pericardial effusion~~[x] and ~~no high-density upper esophageal shadow~~[x].
>
> These two reports are highly similar in wording and structure, yet semantically opposite. Nevertheless, surface-similarity metrics assign high scores (BLEU: 76.5, ROUGE: 80.6, METEOR: 78.4, BioBERTScore: 94.6). In contrast, CABS assigns a score of 0, correctly reflecting the complete mismatch in abnormality sets.
>
> This example illustrates that evaluation hallucination in surface-similarity metrics not only fails to measure clinical correctness, but also provides misleading signals for RL training.
>
> **(Q3) Could the authors provide a clearer description of the CABS calculation pipeline?**
>
> (A3) We thank the reviewer for pointing out the lack of clarity in the CABS definition and computation pipeline. We provide a more detailed explanation here.
>
> At a high level, CABS maps unstructured reports into a structured clinical semantic space, and performs alignment-based evaluation within this space. The core pipeline consists of three steps:
> (1) Parsing: the report is decomposed into a set of abnormal entities, each represented as a structured tuple (abnormality type, location, attributes, etc.);
> (2) Matching: entity-level alignment is performed between the reference and generated reports, based on abnormality type consistency, with additional constraints on location and attributes;
> (3) Scoring: instance-level metrics are computed from matched entities and aggregated into sample-level and benchmark-level metrics.
>
> Instance-level scores serve a dual role: they act as unit rewards for trajectory optimization (used in RL advantage estimation), and as basic units for final evaluation statistics.
>
> The key idea is to shift evaluation from text similarity to clinical semantic unit alignment, explicitly modeling whether abnormalities are correctly identified and described.
>
> Different CABS sub-metrics correspond to different levels of this alignment:
>
> * Entity Core measures whether abnormal entities are correctly identified (precision/recall/F1);
> * Clinical Fidelity further enforces correctness of location and attributes, distinguishing "detected but incorrectly described" cases;
> * Organ Coverage evaluates whether abnormalities are comprehensively covered across organs, capturing biases toward certain regions.
>
> These metrics form a hierarchical progression—from detecting abnormalities, to correctly describing them, to ensuring comprehensive coverage—jointly providing a structured characterization of clinical semantic consistency.
>
> We have added the complete computation pipeline in the revised version (including input/output examples and formulas), along with a workflow diagram and case studies to improve clarity and reproducibility.

---

> > ### Author Rebuttal · Reviewer_MZDZ · 2026-04-02
> >
> > I thank the authors for their responses and for the additional experiments addressing the questions raised, especially the rank comparison experiment. However, I believe the study would benefit from evaluation on a broader set of models, as surface similarity appears to have failed only for GPT-5's rank.
> >
> > Since most of my concerns have been addressed, I will therefore raise my score to weak accept: 4.

---

> > > ### Author Response · Authors · 2026-04-02
> > >
> > > We sincerely thank the reviewer for the positive feedback on our rebuttal. We are very glad to see that the reviewer finds the additional experiments and clarifications helpful, and that most of the concerns have been addressed. We especially appreciate the reviewer’s constructive suggestions, which have significantly helped us improve the quality and completeness of our work.
> > >
> > > We also note that the reviewer mentioned raising the score to **weak accept: 4**. If applicable, **we would appreciate it if the score could be updated in the system,** as this may sometimes require a manual confirmation step.
> > >
> > > Regarding the reviewer’s additional suggestion on evaluating a broader set of models: we fully agree that this is important. Due to space limitations in the rebuttal, we originally reported a subset of ranking comparisons. Here, we provide the complete ranking results across all evaluated models, and further include additional state-of-the-art models, which strengthens our claim:
> > >
> > > | Model | CABS | Surface Similarity | Diff |
> > > |------|:------:|:------:|:-:|
> > > | TIF-GRPO (ours) | 1 | 1 |-|
> > > | GPT-5 | 2 | 10 |+8|
> > > | CT-CHAT | 3 | 2 |-1|
> > > | Gemma 4 26B A4B | 4 | 8 |+4|
> > > | Gemini-3-Pro | 5 | 6 |+1|
> > > | Hulu-Med | 6 | 9 | +3|
> > > | Qwen3-VL-8B | 7 | 7 | -|
> > > | Qwen3.5-27B | 8 | 4 |-4|
> > > | InternVL3.5-8B | 9 | 5 |-4|
> > > | M3D-LaMed | 10 | 11 |+1|
> > > | RadFM | 11 | 12 |+1|
> > > | Fleming-VL-8B | 12 | 3 |-9|
> > >
> > > To further highlight this phenomenon, we summarize several representative cases with significant ranking discrepancies:
> > >
> > > * GPT-5: rank shifts from **2 → 10**
> > > * Gemma 4 26B A4B: rank shifts from **4 → 8**
> > > * Hulu-Med: rank shifts from **6 → 9**
> > > * Qwen3.5-27B: rank shifts from **8 → 4**
> > > * InternVL3.5-8B: rank shifts from **9 → 5**
> > > * Fleming-VL-8B: rank shifts from **12 → 3**
> > >
> > > Notably, these discrepancies occur across both strong general-purpose models and medical-specific models, and include both **overestimation and underestimation** under surface-similarity metrics. This suggests that the issue is not limited to a single model (e.g., GPT-5), but is instead systematic and widespread.
> > >
> > > Once again, we truly appreciate the reviewer’s thoughtful feedback and support.

---

### Official Review · Reviewer_WzFG · 2026-03-11

**Soundness:** 2
**Presentation:** 3
**Significance:** 3
**Originality:** 2
**Overall Recommendation:** 4
**Confidence:** 2

**Summary:**

The paper introduces the Clinical Abnormality Benchmarking Substrate , a structured evaluation system designed to decompose radiology reports into verifiable clinical semantic units. To address the mechanistic divergence where medical vision-language models optimize for surface-level linguistic fluency over factual correctness, the authors propose TIF-GRPO. This framework applies control-theoretic principles to penalize diagnostic omissions and suppress hallucinations iteratively, achieving strong performance improvements on multiple 3D CT benchmarks.

**Compliance With Llm Reviewing Policy:**

Affirmed.

**Final Justification:**

Most of my concerns have been addressed. I will maintain my score at 4.

**Key Questions For Authors:**

Questions

- What is the recall of the CABS extraction process relative to ground-truth abnormalities annotated by radiologists? What fraction of clinician-identified abnormalities are successfully extracted by CABS? If CABS systematically misses certain pathology types, the CABS-based SOTA claims may reflect extraction artifacts rather than genuine clinical improvement.

- How is the canonical ordering of the K reference units determined? Since the model generates tokens sequentially, how is the per-unit hit reward $r_i$ computed incrementally during generation for the running cost calculation?

- The integral feedback mechanism is specifically motivated by the need for "anatomy-aware" optimization, yet the model trained with TIF-GRPO regresses on location accuracy relative to a model trained on more data with standard methods. Can the authors provide a diagnosis of this regression? Does increasing the $\gamma$ weight improve location accuracy?

- The mechanistic divergence experiment constructs "clinically plausible variants via controlled perturbations" of GT reports and establishes a Text-Rank based on "clinical priority order." How are the controlled perturbations generated?

**Limitations:**

yes

**Strengths And Weaknesses:**

Strengths

- The formulation of clinical reasoning as a pseudo-temporal trajectory to calculate rewards is a creative. The CABS framework is clinically well-motivated.

- The analysis of "Mechanistic Divergence" is a highly insightful contribution.

Weaknesses

- The running cost term in Eq. 4 sums over k=1,...,K clinical semantic units in the reference set, treating this as a generative trajectory. However, the model's actual generation is a token sequence, not a sequence of clinical unit predictions. The paper does not specify how the ordering of clinical units in the reference trajectory is determined. Is it the order they appear in the GT report? If the ordering is arbitrary, the running cost's emphasis on "early and persistent misses" is ill-defined.

- TIF-GRPO is evaluated exclusively on 3D CT report generation, a highly specialized domain with limited public data. It is unclear whether the CABS framework and TIF reward design generalize to other medical imaging modalities or to other clinical NLP tasks.

- The comparison baseline set omits concurrent methods that also address clinical factuality in medical VLMs. Without these comparisons, it is unclear whether TIF-GRPO's advantage is primarily from the integral feedback structure or from the clinical semantic decomposition.

---

> ### Author Rebuttal · Authors · 2026-03-31
>
> We appreciate the reviewers for their careful and constructive feedback on our work. Our detailed responses are provided below.
>
> **(Q1) Canonical ordering of K reference units**
>
> (A1) We thank the reviewer for the insightful question. We define a reference-based canonical trajectory, not a token-level generation path. Clinical units are first extracted from the GT report and indexed by their original order, serving only for reward aggregation without constraining generation order.
>
> "Early and persistent misses" arise from this trajectory: earlier anomalies typically correspond to more critical findings and incur larger cumulative penalties, discouraging premature normal descriptions. We verify this design: (a) replacing running cost with order-agnostic averaging degrades performance; (b) forcing "normal" at the first key anomaly in 200 multi-abnormal samples (>3) further reduces Recall (34.6 → 29.7).
>
> |Method|P|R|F1|
> |-|-|-|-|
> |disorderly|38.5|33.9|36.1|
> |orderly|40.9|37.8|39.1|
>
> **(Q2) Recall of CABS extraction vs. radiologist ground truth**
>
> (A2) We thank the reviewer for the suggestion. We randomly sampled 50 reports each from CT-RATE and AMOS-MM, annotated and cross-validated by three experts. We then compared these annotations with outputs from CABS and an additional expert extractor, followed by independent evaluation using Gemini-3-Pro:
>
> |Extractor|P|R|F1|
> |-|-|-|-|
> |Human Expert|96.4|95.7|96.0|
> |CABS|99.2|97.8|98.5|
>
> To further validate CABS against human annotations, we design two additional independent evaluation protocols on the evaluation side. Due to space constraints, please refer to `Reviewer NqCX (A3)`.
>
> **(Q3) Lower Location Accuracy**
>
> (A3) The slight decrease compared to CT-CHAT reflects differences in optimization objectives rather than a degradation in capability. CT-CHAT benefits from large-scale SFT with explicit location supervision (including an additional 700k VQA samples), which enables stronger alignment between abnormalities and their spatial locations.
> In contrast, TIF-GRPO optimizes anomaly detection, consistency, and coverage via report-level structured rewards without extra annotations, prioritizing FN/FP correction.
>
> This is reflected in higher entity-level F1, indicating better overall abnormality recognition. Notably, trajectory weights ($\alpha,\gamma$) control FN/FP trade-offs, not location/attribute trade-offs.
>
> **(Q4) Evaluation scope (3D CT)**
>
> (A4) 3D CT is chosen as a stringent setting (3D reasoning, long-form generation, multi-abnormal consistency). Gains here suggest robustness of the optimization mechanism. TIF-GRPO itself is modality-agnostic, decomposing generation into semantic subgoals with trajectory-level credit assignment.
>
> Cross-modal validation on MIMIC-CXR (2D X-ray) shows consistent improvements:
>
> |Method|P|R|F1|
> |-|-|-|-|
> |Baseline(Qwen3-VL-4B)|21.1|33.1|25.3|
> |SFT|44.1|21.2|28.6|
> |GRPO+ROUGE|52.6|39.3|45.3|
> |GRPO+LLM|60.9|46.7|52.7|
> |TIF-GRPO|74.7|67.5|70.8|
>
> **(Q5) Decomposition vs. integral feedback**
>
> (A5) We thank the reviewer for the concern regarding baseline coverage. We conduct two sets of experiments corresponding to the reviewer’s points on integral feedback structure and clinical semantic decomposition.
>
> We first validate the effectiveness of integral feedback in `A1`.
>
> To evaluate clinical semantic decomposition, we compare against methods based on clinical factuality decomposition without trajectory-level control (OraPO / MRG-R1). These approaches also focus on clinical factuality, but map generated reports into predefined label spaces (via LLM extraction or classification) and optimize at the label level. In contrast, our method directly evaluates generated outputs at the multi-dimensional entity level.
>
> Under aligned experimental settings, TIF-GRPO still significantly outperforms these methods:
>
> |Method(CT-RATE)|P|R|F1|Loc-Acc|Att-Acc|
> |-|-|-|-|-|-|
> |OraPO$^1$|37.0|31.3|33.9|68.2|62.0|
> |MRG-R1$^2$|36.7|33.1|34.8|69.0|63.2|
> |TIF-GRPO|40.9|37.8|39.1|70.2|65.8|
>
> Based on these results, we attribute the gains as follows: clinical semantic decomposition provides higher-fidelity clinical atomic semantic supervision, while integral feedback further improves credit assignment and convergence in multi-objective optimization. Together, they jointly contribute to the final performance gains.
>
> **(Q6) Mechanistic Divergence perturbations**
>
> (A6) Starting from real multi-abnormal reports, we apply template-based, single-variable perturbations on CABS structures: flipping abnormal→normal (miss), replacing with another plausible abnormality (false positive), or modifying attribute/location. Each step introduces one local change, forming a monotonic deviation set.
>
> Under this controlled setup, metrics are fairly compared on ranking fine-grained clinical deviations. Surface-similarity metrics fail to distinguish such structured changes, while CABS-based evaluation better aligns with clinical priorities.

---

> > ### Author Rebuttal · Reviewer_WzFG · 2026-04-03
> >
> > Most of my concerns have been addressed. I will maintain my score at 4.

---

> > > ### Author Response · Authors · 2026-04-03
> > >
> > > We sincerely thank the reviewer for the positive feedback and for acknowledging that most of the concerns have been addressed. We are glad that our rebuttal has been helpful in clarifying the contributions and improving the overall understanding of our work.
> > >
> > > We fully respect the reviewer’s decision to maintain the current score. At the same time, if our responses have successfully resolved the majority of the concerns and strengthened the reviewer’s confidence in the quality and significance of our work, we would like to kindly ask whether it would be possible to consider a slight increase in the confidence level.
> > >
> > > We truly appreciate the reviewer’s time, effort, and thoughtful evaluation of our work.

---

### Official Review · Reviewer_NqCX · 2026-03-11

**Soundness:** 2
**Presentation:** 3
**Significance:** 2
**Originality:** 2
**Overall Recommendation:** 3
**Confidence:** 3

**Summary:**

The paper addresses the misalignment between the RL optimization objectives and the requirements of clinical factuality in the context of Medica-VLMs. It argues that lexical-based proxy signals mostly induce a phenomenon called evaluation hallucination (EH), where models prioritize linguistic fluency over diagnostic precision, also it misguides reward. In EH, a model may achieve state-of-the-art scores on standard benchmarks while committing fatal clinical errors.  To address these problems, the authors mainly introduce:  a) CABS, a structured semantic framework designed to replace unstructured text matching b) TIF-GRPO, a GRPO-based RL framework that shapes rewards using a control-inspired trajectory-integral objective.

**Compliance With Llm Reviewing Policy:**

Affirmed.

**Key Questions For Authors:**

- Why does the hit reward formulation in Eq. 3 drop the attribute A and certainty c components defined in the CABS in Eq. 2? How does the TIF-GRPO framework ensure the model learns to accurately report critical details like lesion size or severity if they are excluded from the direct RL optimization signal?

- Could you provide performance metrics (mean and std) averaged across multiple random seeds? How much the results from independent runs matching with radiologist scores? What are confidence intervals and p-values?

- What is the standalone, quantitative error rate of the Stage-1 CABS extractor when evaluated against human-labeled standard? How can researchers be confident that the policy model is not simply overfitting to the specific biases and blind spots of the extraction LLM?

**Limitations:**

The method is only as faithful as its structured decomposition and scoring.  If model mis-parses a report or if the underlying report is itself incomplete/biased relative to the image, then optimizing against extracted abnormality units can incentivize “agreement with the report style” rather than “agreement with ground-truth imaging".

**Strengths And Weaknesses:**

Strength:

- The paper maps the text generation process to a control system by formulating diagnostic omissions as steady-state errors and hallucinations as excessive control effort.

- The experimental design includes a diverse set of baselines, including one of the giants, GPT-5 and medical-specific models like RadFM and CT-CHAT.

- Starting from GRPO, the paper defines a CABS-based reward at the abnormality-unit level and then introduces a trajectory-integral reward. This composite reward is novel in this domain.

Weaknesses

- The paper designs the CABS framework to capture clinical structure,  as defined in Eq. 2. However, the actual RL reward function defined in Eq. 3 utilizes the binary indicators for the entity d and location l. In radiology, the difference between a vague "liver lesion" and a "X cm lesion in segment Y (or suspicious for Y)" is clinically important. Because, even if the reward calculation drops these attributes, the model receives the exact same optimization signal, whether it gets the size, morphology, or certainty right or wrong. This creates a mismatch between the representational richness of the metric and the actual training signal.

- The authors explicitly motivate the methodology by stating that standard RL facing instability problem in this domain.  However, the experimental results provided in tables report only single-point estimates. Without reporting results across multiple random seeds, including means, standard deviations, confidence intervals, it is impossible to statistically verify whether the observed improvements are robust or merely the results of lucky seed.

- The paper does not provide a quantitative metric for how often the Stage-1 LLM extractor produces incorrect outputs. While the authors report 98.6% radiologist approval rate , this may rely on a subjective scoring and does not explicitly separate the extractor's hallucination rate from its miss rate. Consequently, the policy model is optimizing for "what the extractor can see," not necessarily what is actually in the given image. So, the reported sota performance may simply reflect the policy model learning to mimic the specific extraction artifacts, biases, or systematic errors of the CABS agent.

---

> ### Author Rebuttal · Authors · 2026-03-31
>
> We sincerely thank the reviewers for their thoughtful assessment of our work. We provide detailed responses below.
>
> **(Q1) Why does Eq. 3 use only entity and location indicators?**
>
> (A1) We thank the reviewer for the insightful question regarding the CABS reward. In fact, we deliberately distinguish between reward signal and evaluation: during evaluation, all attributes are explicitly considered, whereas in RL optimization we retain only entity and location signals. This is because these two types of signals are more stable and discriminative, making them better suited for forming consistent ranking criteria under group-relative optimization.
>
> In contrast, attribute and certainty are more prone to noise and harder to align. As noted by the reviewer, fine-grained numerical details are more susceptible to irreversible deviations during data preprocessing and semantic mapping, which can weaken the accumulative error structure required by the trajectory-integral reward. (At the evaluation stage, only qualitative attribute descriptions are retained to ensure a fair and unbiased comparison.)
>
> Nevertheless, attribute-related representations are already learned during the SFT stage and are preserved during RL via KL regularization. We also empirically tested incorporating attribute or certainty directly into the reward. These variants yield only marginal improvements in attribute-related metrics, while significantly diluting the overall optimization objective and reducing the final F1 score:
>
> |Reward|Loc-Acc|Att-Acc|F1|
> |-|-|-|-|
> |TIF-GRPO|70.2|65.8|39.1|
> |+attribute|-1.0|+4.8|-1.6|
> |+certainty|-0.5|-1.3|-0.2|
> |+both|-1.1|+5.3|-1.4|
>
> **(Q2) Are the reported gains statistically robust across random seeds?**
>
> (A2) We thank the reviewer for emphasizing statistical rigor. In practice, our method demonstrates consistent optimization trends across datasets (Table 1,2), task formats (Table 3), model scales (Table 1,2), and multiple ablation settings (Table 4,5), indicating stable optimization behavior.
>
> Furthermore, we provide additional experiments with three random seeds (seed = 0, 42, 3407). For each seed, we perform inference three times with temperature 0.7, and initialize RL training from SFT checkpoints with median performance. We also conduct two-sided Welch’s t-tests against GRPO + ROUGE and construct confidence intervals based on the t-distribution.
>
> The results show low variance across key metrics, and the advantage of TIF-GRPO over major baselines remains consistent across random seeds:
>
> |Method|Mean ± SD|95% CI|p-value|
> |-|-|-|-|
> |SFT|18.40 ± 1.73|[14.10,22.70]|-|
> |GRPO+ROUGE|29.54 ± 2.82|[22.54,36.54]|-|
> |GRPO+RadGraph|25.32 ± 1.48|[21.63,29.01]|0.04877|
> |GRPO+BioBERT|28.03 ± 2.53|[21.74,34.32]|0.37535|
> |GRPO+LLM|31.56 ± 2.35|[25.73,37.40]|0.16079|
> |TIF-GRPO|40.20 ± 1.22|[37.18,43.23]|0.00825|
>
> **(Q3) How reliable is the Stage-1 CABS extractor, and how do you rule out policy overfitting to extractor bias rather than image-grounded clinical signals?**
>
> (A3) We thank the reviewer for raising concerns about the reliability of the Stage-1 CABS extractor. The reported 98.6% acceptance rate is based on LLM self-evaluation aligned with strict criteria, as well as consistency verified by multiple independent experts (Table 6). Approximately 93.9% of samples exhibit one-to-one correspondence with the original reports at the abnormal-entity level (Clinical Decomposition Usability = 5). The remaining discrepancies mainly arise from differences in entity splitting/merging rather than systematic hallucination or omission.
>
> To directly evaluate the extractor, we randomly sampled 50 reports each from CT-RATE and AMOS-MM, annotated and cross-validated by three experts. We then compared these annotations with outputs from CABS and another expert extractor, followed by independent evaluation using Gemini-3-Pro:
>
> |Extractor|P|R|F1|
> |-|-|-|-|
> |Human Expert|96.4|95.7|96.0|
> |CABS|99.2|97.8|98.5|
>
> These results demonstrate the high reliability of CABS. More direct evidence comes from cross-evaluator consistency. We further provide independent comparisons: four radiologists (2 junior + 2 senior) conducted blind ranking over 200 generated samples, yielding results fully consistent with CABS. Additionally, using CT-RATE classification labels, we computed label-vector distances via a pretrained classification model RadBert, which produced highly consistent rankings. Across all evaluation systems, TIF-GRPO consistently ranks first:
>
> |Model|CABS|Surface Similarity|Expert|RadBert|
> |-|-|-|-|-|
> |TIF-GRPO|1|1|1|1|
> |GPT-5|2|4|2|3|
> |CT-CHAT|3|2|3|2|
> |Hulu-Med|4|3|4|4|
> |M3D-LaMed|5|5|5|5|
> |RadFM|6|6|6|6|
>
> These evaluation protocols are mutually independent yet yield consistent conclusions, indicating that CABS effectively extracts clinically meaningful signals aligned with imaging, rather than introducing bias or systematic errors.

---

### Official Review · Reviewer_1qMk · 2026-03-12

**Soundness:** 3
**Presentation:** 2
**Significance:** 2
**Originality:** 2
**Overall Recommendation:** 4
**Confidence:** 2

**Summary:**

This paper tackles the critical issue of "evaluation hallucinations" and "mechanistic divergence" in 3D Medical Vision-Language Models (VLMs). The authors propose a two-part framework: (1) CABS, an anatomical ontology that decomposes radiology reports into structured, verifiable tuples ($f_i = \langle o, d, \mathcal{A}, l, c, e \rangle$), and (2) TIF-GRPO, an adaptation of Group Relative Policy Optimization that uses CABS to compute integral rewards. By framing the reasoning process as a pseudo-temporal sequence, TIF-GRPO applies running costs to penalize omitted abnormalities and control effort penalties to suppress hallucinations. Experimental results on CT-RATE and AMOS-MM show state-of-the-art performance, significant improvements in clinical fidelity, and high data efficiency compared to baselines like CT-CHAT.

**Compliance With Llm Reviewing Policy:**

Affirmed.

**Final Justification:**

I believe the authors have addressed the major concern of this paper, and I raise my overall score to 4. However, I still have concerns about the method's novelty itself, but I believe the proposed method is solid enough. Hence, I decrease my confidence score.

**Key Questions For Authors:**

See Weakness

**Limitations:**

yes

**Strengths And Weaknesses:**

**Strengths**

- Highly Effective Problem Formulation: The identification of "mechanistic divergence" (where models optimize for linguistic fluency over clinical factuality) is a pressing issue in trustworthy medical AI. The empirical demonstration that standard NLP metrics (BLEU, ROUGE) correlate poorly with clinical accuracy is well-executed and compelling.

- Strong Empirical Results: The performance gains are substantial. Achieving a significant bump in Recall and F1 over domain-specific 8B models using a 4B model and only 45% of the training data demonstrates the high efficiency of the proposed reward signal.

- Robust Evaluation: The ablation studies clearly isolate the impact of different reward components, proving that the specifically tuned $\alpha/\gamma$ ratios successfully govern the Precision/Recall trade-off.

**Weaknesses**

- Limited Algorithmic Innovation: The primary methodological contribution, TIF-GRPO, appears to be an application of standard GRPO augmented with domain-specific reward shaping. The integration of "control-theoretic principles" reads more like a rebranding of cumulative penalty tracking for False Positives and False Negatives over generation steps. The algorithm does not fundamentally advance RL methodology, but rather applies a highly engineered, domain-specific reward function to an existing optimizer.

- Over-reliance on Domain Ontology: The success of the framework is almost entirely bottlenecked by CABS. Because the reward function $R_{TIF}$ relies on the explicit structuring of medical tuples ($f_i$), the "generalizability" claimed by the authors is restricted strictly to domains where such a rigid, atomic ontology exists and can be reliably extracted. This limits its broader impact for the general machine learning community.

- Scalability of the Reward Model: Extracting the CABS tuples to calculate the "semantic fact intersection" during RL training requires either a flawless rule-based parser or a highly capable LLM as an evaluator in the loop. The paper needs to clarify the computational overhead of computing these trajectory-integral rewards during the GRPO rollout phase.

---

> ### Author Rebuttal · Authors · 2026-03-31
>
> We first thank the reviewers for their careful evaluation of our work. Below we provide our detailed responses.
>
> **(Q1) Limited Algorithmic Innovation**
>
> (A1) We thank the reviewer for their attention to methodological novelty. We would like to clarify that **the contribution of this work is not to propose a universal RL algorithm applicable to all scenarios**, but rather to introduce a more suitable framework targeting the two core issues in 3D CT report generation: evaluation hallucination and mechanistic divergence.
>
> Conventional reward shaping typically performs pointwise transformations of local rewards and does not alter the credit assignment structure. In contrast, TIF-GRPO introduces a trajectory-integral reward, transforming the optimization objective from a simple summation of anomaly-unit contributions into a path-dependent objective function. This fundamentally changes the credit assignment across trajectory sequences, leading to structural changes in group-wise advantage ranking and corresponding gradient directions. By incorporating trajectory-level signals, TIF-GRPO optimizes the action policy during RL training in a targeted manner, explicitly steering optimization toward clinically meaningful trade-offs. This distinguishes it from reward shaping methods based purely on numerical adjustments.
>
> Table 4 demonstrates that trajectory-level control is not merely a reparameterization of FN/FP, but provides explicit control over clinical diagnostic preferences. Table 5 further shows that domain-specific reward shaping alone is insufficient. We also compare against concurrent clinical factuality decomposition methods (OraPO/MRG-R1), where TIF-GRPO still maintains superior performance:
>
> |Method(CT-RATE)|P|R|F1|Loc-Acc|Att-Acc|
> |-|-|-|-|-|-|
> |OraPO $^1$|37.0|31.3|33.9|68.2|62.0|
> |MRG-R1 $^2$|36.7|33.1|34.8|69.0|63.2|
> |TIF-GRPO|40.9|37.8|39.1|70.2|65.8|
>
> Therefore, the key innovation of TIF-GRPO lies in redefining the relationship between reward and trajectory, altering the credit assignment mechanism through a trajectory-integral formulation, which leads to significant performance improvements.
>
> **(Q2) Over-reliance on Domain Ontology**
>
> (A2) We thank the reviewer for raising concerns about generalizability. We agree that the current setup relies on the CABS instantiation; however, CABS is a prerequisite of the 3D CT report generation setting rather than a limitation of the method itself. We emphasize that the core of TIF-GRPO does not depend on any specific ontology, but instead on two more general conditions:
> (1) the task can be decomposed into verifiable intermediate semantic units, and
> (2) these units can be incorporated into trajectory-level credit assignment.
>
> Under these conditions, our framework remains applicable. Therefore, our notion of generalizability is not to cover all tasks, but to provide a unified modeling approach for a class of problems that previously lacked effective optimization paradigms.
>
> We further validate this through cross-modal experiments on the MIMIC-CXR dataset, where TIF-GRPO still achieves significant performance gains:
>
> |Method|P|R|F1|
> |-|-|-|-|
> |Baseline(Qwen3-VL-4B)|21.1|33.1|25.3|
> |SFT|44.1|21.2|28.6|
> |GRPO+ROUGE|52.6|39.3|45.3|
> |GRPO+LLM(rubricreward)|60.9|46.7|52.7|
> |TIF-GRPO|74.7|67.5|70.8|
>
> In summary, we position the generalizability of this work as arising from a unified modeling framework for decomposable semantic generation problems, rather than reliance on a specific medical ontology.
>
> **(Q3) Scalability of the Reward Model**
>
> (A3) We thank the reviewer for their concerns regarding reward computation cost and scalability. In practice, we only use a moderately sized model (Qwen3-30B-A3B) to reliably support anomaly entity extraction and evaluation during RL training. Therefore, TIF-GRPO does not require a flawless parser or a high-cost LLM in the rollout phase. The additional overhead mainly comes from the parsing and matching between generated reports and structured reference reports.
>
> Under local deployment (8×H100), the training cost increases by approximately 77.4% compared to surface-level similarity rewards, but is only about 6% higher than general LLM-as-judge (rubric-based) rewards. Under API parallelization, training latency increases by about 8%, corresponding to a cost on the order of $10^{-4}$ per sample. This overhead primarily yields higher-quality, more discriminative, and more stable optimization signals, rather than increased model complexity.
>
> More importantly, this cost is well compensated by improvements in sample efficiency. TIF-GRPO, using only 10% of the data (18.3% cost of GRPO-ROUGE), already achieves performance close to GRPO-ROUGE; and with 50% of the data (88.2% cost), it significantly surpasses it (F1: 34.8 vs. 29.7). This demonstrates that the method achieves better performance with less data, improving overall training efficiency while reducing cost.
>
> References
>
> [1] Chen et al., OraPO, 2025
>
> [2] Wang et al., MRG-R1, 2025

---

> > ### Author Rebuttal · Reviewer_1qMk · 2026-04-02
> >
> > The rebuttal satisfactorily addresses my concerns about scalability and partially addresses the concern about generalizability by narrowing the claim to decomposable semantic generation tasks and providing cross-modal evidence. However, my main reservation regarding methodological novelty remains. The current response strengthens the case for empirical effectiveness, but does not yet convincingly establish that TIF-GRPO constitutes a substantive algorithmic advance beyond GRPO equipped with carefully designed, domain-specific trajectory rewards. Therefore, while my confidence in the practical value of the work has increased, my core concern about novelty is only partially resolved.

---

> > > ### Author Response · Authors · 2026-04-02
> > >
> > > We thank the reviewer for recognizing the practical value of our work and are pleased that we have reached greater consensus on scalability and generalizability. We understand that the remaining concern centers on methodological novelty, and we would like to provide further clarification.
> > >
> > > Specifically, although TIF-GRPO builds upon GRPO, its key distinction lies not in simple reward engineering, but in altering the distribution of advantage estimation through a trajectory-level perspective. We model long-form medical report generation as a controllable ""diagnostic trajectory" and regulate this trajectory continuously during optimization, effectively shifting RL from a "reward design problem" to a "trajectory regulation problem," which is fundamentally different from conventional reward shaping. More importantly, this work is the first to systematically identify and empirically validate the phenomenon of "mechanistic divergence," where RL optimization in medical settings can systematically deviate from clinical factuality rather than merely reflecting metric misalignment. Building on this insight, TIF-GRPO not only addresses this issue at the optimization level but also outperforms concurrent clinical factuality decomposition methods such as OraPO and MRG-R1, forming a complete contribution loop of "problem identification – mechanism analysis – optimization redesign."
> > >
> > >
> > > We also understand the reviewer’s expectation for novelty from the perspective of general RL algorithm development. At the same time, since the primary area of this work is **Applications → Health / Medicine,** we would kindly ask the reviewer to also consider the contribution from the perspective of addressing critical domain-specific challenges. From this viewpoint, the significance of TIF-GRPO lies in redefining the RL optimization paradigm for medical report generation—from surface-level similarity optimization to clinically grounded trajectory-level regulation—and we have demonstrated its consistent effectiveness across multiple tasks and modalities (`Tables 1–3` and answer `(A2)`).
> > >
> > >
> > > Regarding generalizability, we emphasize that its generalizability is targeted toward a class of problems characterized by "decomposable and verifiable" long-form generation. Such problems are central and widely present in medical and other high-reliability domains. Therefore, we position TIF-GRPO  as a unified optimization framework for this class of problems.
> > >
> > > In summary, we fully respect the reviewer’s careful assessment regarding novelty. However, we sincerely hope the reviewer can also consider that this work (1) identifies and validates the critical issue of mechanistic divergence, (2) reconstructs the RL optimization objective and credit assignment mechanism accordingly, and (3) provides a stable, reusable, and significantly more effective solution in the medical domain. From an application-driven and problem-oriented perspective, we believe this work offers substantial research value. If the reviewer finds this perspective reasonable, we would sincerely appreciate consideration for a score improvement. We thank the reviewer again for the constructive feedback.

---

### Decision · Program_Chairs · 2026-04-30

**Decision:**

Accept (regular)

**Comment:**

Reviewers agreed that the paper features a highly effective problem formulation by identifying "mechanistic divergence" in trustworthy medical AI, presents strong empirical results with substantial performance gains in recall and F1 using a more efficient model and less training data, conducts robust ablation studies, and offers creative contributions including formulating clinical reasoning as a pseudo-temporal trajectory and the clinically well-motivated CABS framework, which is novel in the domain and provides valuable insights into VLM evaluation. However, they also raised serious concerns about the paper's soundness and generalizability: the validation of the CABS framework’s effectiveness is insufficient and the method’s generalizability is limited by its heavy dependence on domain ontology and exclusive evaluation on 3D CT report generation, with unclear scalability of the reward model. The rebuttal addressed most of the above concerns and two reviewers raised their scores. But, there is still no unanimous agreement among reviewers. After careful deliberation, I recommend accepting this paper since its merits over-weights its limitations.